# Human Embryonic Stem Cell-Derived Immature Midbrain Dopaminergic Neurons Transplanted in Parkinsonian Monkeys

**DOI:** 10.3390/cells12232738

**Published:** 2023-11-30

**Authors:** Adolfo López-Ornelas, Itzel Escobedo-Avila, Gabriel Ramírez-García, Rolando Lara-Rodarte, César Meléndez-Ramírez, Beetsi Urrieta-Chávez, Tonatiuh Barrios-García, Verónica A. Cáceres-Chávez, Xóchitl Flores-Ponce, Francia Carmona, Carlos Alberto Reynoso, Carlos Aguilar, Nora E. Kerik, Luisa Rocha, Leticia Verdugo-Díaz, Víctor Treviño, José Bargas, Verónica Ramos-Mejía, Juan Fernández-Ruiz, Aurelio Campos-Romo, Iván Velasco

**Affiliations:** 1Instituto de Fisiología Celular—Neurociencias, Universidad Nacional Autónoma de México, Mexico City 04510, Mexico; adolfolopezmd@gmail.com (A.L.-O.); iescobed@ifc.unam.mx (I.E.-A.); jrolandolara@gmail.com (R.L.-R.); cmelendez@ifc.unam.mx (C.M.-R.); beetsi@ciencias.unam.mx (B.U.-C.); acaceres@ciencias.unam.mx (V.A.C.-C.); xochitlflores91@gmail.com (X.F.-P.); jbargas@ifc.unam.mx (J.B.); 2Laboratorio de Reprogramación Celular, Instituto Nacional de Neurología y Neurocirugía Manuel Velasco Suárez, Mexico City 14269, Mexico; 3División de Investigación, Hospital Juárez de México, Mexico City 07760, Mexico; 4Departamento de Fisiología, Facultad de Medicina, Universidad Nacional Autónoma de México, Mexico City 04510, Mexico; leticia@unam.mx (L.V.-D.); jfr@unam.mx (J.F.-R.); 5Unidad Periférica de Neurociencias, Facultad de Medicina, Universidad Nacional Autónoma de México, Instituto Nacional de Neurología y Neurocirugía Manuel Velasco Suárez, Mexico City 14269, Mexico; gabbo_fry@hotmail.com; 6Escuela de Medicina y Ciencias de la Salud, Tecnológico de Monterrey, Monterrey 64710, Mexico; tonabarrios@yahoo.com (T.B.-G.); vtrevino@tec.mx (V.T.); 7Departamento de Farmacobiología, Centro de Investigación y de Estudios Avanzados del IPN (Cinvestav), Mexico City 07360, Mexico; fcarmona@cinvestav.mx (F.C.); lrocha@cinvestav.mx (L.R.); 8Molecular Imaging PET-CT Unit, Instituto Nacional de Neurología y Neurocirugía Manuel Velasco Suárez, Mexico City 14269, Mexico; betoven@ciencias.unam.mx (C.A.R.); capagp@gmail.com (C.A.); nora.kerik@hotmail.com (N.E.K.); 9Gene Regulation, Stem Cells, and Development Group, GENYO-Centre for Genomics and Oncological Research Pfizer, University of Granada, Andalusian Regional Government, PTS, 18016 Granada, Spain; veronica.ramos@genyo.es

**Keywords:** brain grafting, cell therapy, depolarization-induced dopamine release, PET, MRI

## Abstract

Human embryonic stem cells (hESCs) differentiate into specialized cells, including midbrain dopaminergic neurons (DANs), and Non-human primates (NHPs) injected with 1-methyl-4-phenyl-1,2,3,6-tetrahydropyridine develop some alterations observed in Parkinson’s disease (PD) patients. Here, we obtained well-characterized DANs from hESCs and transplanted them into two parkinsonian monkeys to assess their behavioral and imaging changes. DANs from hESCs expressed dopaminergic markers, generated action potentials, and released dopamine (DA) in vitro. These neurons were transplanted bilaterally into the putamen of parkinsonian NHPs, and using magnetic resonance imaging techniques, we calculated the fractional anisotropy (FA) and mean diffusivity (MD), both employed for the first time for these purposes, to detect in vivo axonal and cellular density changes in the brain. Likewise, positron-emission tomography scans were performed to evaluate grafted DANs. Histological analyses identified grafted DANs, which were quantified stereologically. After grafting, animals showed signs of partially improved motor behavior in some of the HALLWAY motor tasks. Improvement in motor evaluations was inversely correlated with increases in bilateral FA. MD did not correlate with behavior but presented a negative correlation with FA. We also found higher 11C-DTBZ binding in positron-emission tomography scans associated with grafts. Higher DA levels measured by microdialysis after stimulation with a high-potassium solution or amphetamine were present in grafted animals after ten months, which has not been previously reported. Postmortem analysis of NHP brains showed that transplanted DANs survived in the putamen long-term, without developing tumors, in immunosuppressed animals. Although these results need to be confirmed with larger groups of NHPs, our molecular, behavioral, biochemical, and imaging findings support the integration and survival of human DANs in this pre-clinical PD model.

## 1. Introduction

Parkinson’s disease (PD) is a neurodegenerative and progressive disorder caused by alterations in the basal ganglia circuitry after the degeneration of dopaminergic neurons (DANs) in the substantia nigra pars compacta, leading to a reduction of dopamine (DA) levels in the striatum. Clinically, PD is characterized by resting tremor, rigidity, and bradykinesia [1,2]. The brains of non-human primates (NHPs) are anatomically and physiologically like the human brain. The mitochondrial complex I inhibitor 1-methyl-4-phenyl-1,2,3,6-tetrahydropyridine (MPTP) is a neurotoxin employed to develop NHP PD models because it reproduces some histological and motor alterations [3,4,5].

To provide therapeutic strategies, human pluripotent cells, such as embryonic stem cells (hESCs) and induced pluripotent stem cells, are used to generate DANs in vitro [6,7,8,9] by exposing them to molecules that stimulate signaling pathways found in ventral midbrain development. Mouse and human pluripotent stem cell-derived DANs, grafted into the striatum of 6-hydroxy-dopamine-lesioned rodents, exhibit long-term survival and improvements in rotation behavior and akinesia [10,11,12,13].

A protocol for the differentiation of human pluripotent stem cells that produces homogeneous cultures of floor-plate-derived mesencephalic DANs was used to graft MPTP-treated monkeys, and these neurons survived for a month without tumor formation [12]. This method, and others, have been used to promote the differentiation of induced pluripotent stem cells to DANs which, after grafting, improved the behavioral alterations present in parkinsonian NHPs [14,15,16,17] and stabilized symptoms in one PD patient [18]. Although behavioral recovery has been observed post-grafting in parkinsonian NHPs, there are no reports of DA release in the brains of transplanted animals.

Currently, imaging studies such as magnetic resonance imaging (MRI) and positron emission tomography (PET) assess structural and functional changes in the brain. A long-lasting recovery of dopaminergic sites detected by PET has been reported after grafting DANs in the brain of NHPs [14,16], but changes detected in MRI post-grafting have not been analyzed. Here we assessed the therapeutic viability of differentiated DANs to recover the alterations present in parkinsonian NHPs. We used molecular, behavioral, biochemical, and imaging techniques to comprehensively study the impact of the graft and found significant improvements associated with the presence of DANs in the putamen after ten months. Importantly, the brains of grafted NHPs presented MRI changes correlated with behavior; higher DA release was also detected in vivo. This study reinforced the notion that cell replacement therapy might effectively treat PD.

## 2. Results

### 2.1. Dopaminergic Neuron Differentiation

Enhanced Green fluorescent protein (EGFP) expressing hESCs were employed [19]. We first corroborated the expression of pluripotency markers in H9 wild-type hESCS by detecting OCT4, SOX2, NANOG, and SSEA4 (Appendix A). Subsequently, the H9-EGFP cell line was also confirmed to express *SOX2*, *OCT4*, and *KLF4* (Appendix A), to present a euploid karyotype (Appendix A), and to produce teratomas after inoculation in nu/nu immunodeficient mice (Appendix A). Differentiation to midbrain floor-plate-derived DANs was induced, as reported [12] in H9-EGFP cells. We confirmed the expression of relevant ventral midbrain (*FOXA2*) and dopaminergic (*LMX1A*) markers in agreement with a previous study [12], suggesting that our differentiation was successful. *LMX1A* and *FOXA2* were present from day 7 of differentiation and significantly increased on days 21–38, and the expression of *Tyrosine Hydroxylase (TH)* was detected from day 21 and significantly elevated on days 31–38 (Figure 1A). RNA-Seq at different differentiation times revealed that pluripotency genes were down-regulated as cell commitment proceeded with concomitant induction of DAN genes. These DA-related markers included, among others: *DDC*, *FOXA1*, *TH*, *LMX1A*, *HES6*, *PAX6*, *NR4A2*, *DCX*, *SYT4*, *NEUROG2*, *ASCL1*, and *GLI3* (Figure 1B, green arrows) at days 14 and 28. The expression of *FOXA2* at day 0 of the differentiation protocol was undetected (0 FPKM), increased at day 14 (26.5 ± 9 FPKM), and decreased at day 28 (6.5 ± 2.5 FPKM) of the protocol. A similar expression pattern is observed in 2D platforms during the transition of mature DANs [20]. *FOXA1*, another ventral midbrain marker was increased through the protocol differentiation, as expected. Both factors are involved in the development, specification, and maturation of the physiological functions of DANs [21]. On the other hand, the pluripotency markers *NANOG*, *LIN28A*, *LIN28B*, and *POU5F1* (Figure 1B, red arrows) notably reduced their expression on these days compared to day 0. A comparison of the microarray data presented by Kriks and colleagues [12] with selected transcripts from our RNA-Seq results is presented in Appendix A. The full analysis of the sequencing data is presented elsewhere [22].

### 2.2. Differentiated Dopaminergic Neurons Have a Mature Phenotype In Vitro

The expression of the neural precursor marker NESTIN was confirmed during the differentiation protocol on day 14 (Figure 1C). On day 35, the expression of the ventral midbrain DAN marker (FOXA2), DAN marker (TH), and neuronal marker (βIII-TUBULIN) were detected (Figure 1D,E). Quantitative co-expression analysis for FOXA2/βIII-tubulin and TH/βIII-tubulin showed progressive and significant increases in double-positive cells, reaching 86% and 76% at day 42, respectively (Figure 1F), similar to previous work [12]. To analyze if mature neurons expressing MAP2 were also positive for FOXA2, immunolabeling with these markers showed that 72.2% were double-positive at day 38 (Appendix A). To further characterize cell phenotypes present in differentiating cultures at day 21, we found that 16% of the total cells expressed GFAP (astrocytic marker) and the remaining EGFP cells were positive to βIII-TUBULIN (Appendix A); apart from DANs, we found other neuronal populations that expressed GAD65/67 (GABAergic marker, 11.4%) and Serotonin (2.4%) (Appendix A). These results indicate that the predominant phenotype is DANs with appropriate midbrain identity. Electrophysiological recordings of differentiated cells between days 50 and 62 showed spontaneous single spike and bursting activity (6/10 neurons; Figure 1G; top 2 rows) characteristic of DAN identity. Furthermore, all cells (*n* = 10) elicited action potentials induced by current pulses (Figure 1G, third and fourth rows). The neurons also showed rebound spikes in response to hyperpolarizing current pulses (*n* = 6), rectification currents (*n* = 8), and spontaneous synaptic activity (*n* = 6). The shape and duration of an action potential are presented in Figure 1G. It is worth mentioning that recorded neurons showed both mature and immature electrophysiological features of DANs. The mature signs were depolarization block at high stimulus intensities, autonomous firing at low frequencies (1 to 3 Hz), a firing threshold of −43 ± 3 mV, a membrane potential at zero current of −54 ± 4 mV, and an action potential amplitude of 57 ± 1.7 mV (*n* = 5). However, immature characteristics persist after 55–65 days of differentiation, including spontaneous action potentials with a duration of 3.7 ± 0.22 ms measured at half-width, compared with 2.3 ± 0.1 ms in more mature DANs from rats [23]. However, after hyperpolarizing (AHPs), the potentials were not fully developed, perhaps because K+ currents are not completely expressed [24]. Finally, a strong inward rectification due to HCN channel current, commonly seen as a “sag” during evoked hyperpolarization, was not observed. On day 70, the cultures released DA, measured by HPLC, when stimulated with a high potassium medium (Figure 1H). Thus, this differentiation procedure generated mature DANs.

The expression of the neural precursor marker NESTIN was confirmed during the differentiation protocol on day 14 (Figure 1C). On day 35, the expression of the ventral midbrain DAN marker (FOXA2), DAN marker (TH), and neuronal marker (βIII-TUBULIN) were detected (Figure 1D,E). Quantitative co-expression analysis for FOXA2/βIII-tubulin and TH/βIII-tubulin showed progressive and significant increases in double-positive cells, reaching 86% and 76% at day 42, respectively (Figure 1F), similar to previous work [12]. To analyze if mature neurons expressing MAP2 were also positive for FOXA2, immunolabeling with these markers showed that 72.2% were double-positive at day 38 (Appendix A). To further characterize cell phenotypes present in differentiating cultures at day 21, we found that 16% of the total cells expressed GFAP (astrocytic marker) and the remaining EGFP cells were positive to βIII-TUBULIN (Appendix A); apart from DANs, we found other neuronal populations that expressed GAD65/67 (GABAergic marker, 11.4%) and Serotonin (2.4%) (Appendix A). These results indicate that the predominant phenotype is DANs with appropriate midbrain identity. Electrophysiological recordings of differentiated cells between days 50 and 62 showed spontaneous single spike and bursting activity (6/10 neurons; Figure 1G; top 2 rows) characteristic of DAN identity. Furthermore, all cells (*n* = 10) elicited action potentials induced by current pulses (Figure 1G, third and fourth rows). The neurons also showed rebound spikes in response to hyperpolarizing current pulses (*n* = 6), rectification currents (*n* = 8), and spontaneous synaptic activity (*n* = 6). The shape and duration of an action potential are presented in Figure 1G. It is worth mentioning that recorded neurons showed both mature and immature electrophysiological features of DANs. The mature signs were depolarization block at high stimulus intensities, autonomous firing at low frequencies (1 to 3 Hz), a firing threshold of −43 ± 3 mV, a membrane potential at zero current of −54 ± 4 mV, and an action potential amplitude of 57 ± 1.7 mV (*n* = 5). However, immature characteristics persist after 55–65 days of differentiation, including spontaneous action potentials with a duration of 3.7 ± 0.22 ms measured at half-width, compared with 2.3 ± 0.1 ms in more mature DANs from rats [23]. However, after hyperpolarizing (AHPs), the potentials were not fully developed, perhaps because K+ currents are not completely expressed [24]. Finally, a strong inward rectification due to HCN channel current, commonly seen as a “sag” during evoked hyperpolarization, was not observed. On day 70, the cultures released DA, measured by HPLC, when stimulated with a high potassium medium (Figure 1H). Thus, this differentiation procedure generated mature DANs.

### 2.3. hESC-Derived Dopaminergic Neurons Promote Behavioral Recovery in MPTP-Treated NHPs

Immature DANs differentiated from hESCs for 22 days were grafted into parkinsonian NHPs, treated with MPTP administered through multiple intramuscular injections (0.5 mg/kg to reach 2–2.5 mg/kg divided into 4 or 5 daily), which developed motor alterations similar to those seen in patients [25]. We performed imaging, behavioral, and biochemical analyses (Figure 2A). Animals were trained to perform the HALLWAY task before intoxication with MPTP, and the effects of grafting hESC-derived DANs were analyzed for 10 months, as previously described [4,5]. To assess ambulatory and fine motor behavior, three successive behaviors, previously validated to be significantly modified after MPTP administration [4], were evaluated: displacement, reaching, and ingestion (Figure 2B). The coordinates for grafting were obtained by MRI for each subject 3 months after MPTP administration to cover the entire putamen (Figure 2C). Before grafting, MPTP administration increased the times that animals used to complete the tasks, with significant increases at least in one test, compared to the basal condition of NHPs (Appendix A and Figure 2D). After 10 months, sham grafting in one NHP did not induce overt behavioral improvement, supporting the notion that the lesion was permanent.

In contrast, Grafted 1 exhibited significantly decreased performance times lasting 10 months. Grafted 2 showed significant improvements compared to MPTP during the first 4 months, with variable recovery afterward (Figure 2D). Consistent with these differences, Grafted 1 showed a significant decrease in the number of ingested rewards after MPTP, especially on shelf 4 and the well, which was recovered after grafting. A similar decline after MPTP and recovery post-grafting in hallway crossings was observed for Grafted 1 (mean ± SEM): pre-MPTP, 14.4 ± 0.7; MPTP, 3.8 ± 0.6; 10-months average post-grafting, 12.5 ± 0.3, indicating that this NHP was the most affected in its motor activity after MPTP and that grafting of DANs partially recovered in the HALLWAY motor tasks. The total performance time for the complete hallway test did not present significant differences among subjects. We performed a single-case design analysis, consisting of comparing the 10-month average times in relation to MPTP time (Appendix A), for each behavioral task. We observed that the Sham NHP marginally decreased the average time for reaching and ingestion in the 10 months after surgery when compared to MPTP. Grafted 1 showed improved average times in all behavioral tasks, while Grafted 2 showed decreased times in reaching and ingestion in the 10 months post-grafting. Statistical analysis showed that Sham and Grafted 2 were significantly different from Grafted 1 in displacement. For reaching, the three NHPs were significantly different from each other. Ingestion showed a significant recovery of both grafted NHPs when compared to the sham (Figure 2D, bottom part). These data suggest that bilateral DAN transplantation induces behavioral recovery, especially in ingestion in both grafted NHPs.

### 2.4. Diffusion Tensor Imaging

We performed fractional anisotropy (FA) and mean diffusivity (MD) analyses to assess cellular and axonal density changes by MRI. Before grafting, we analyzed the overall changes in the left and right putamen of three healthy NHPs, different from those used for sham/grafting surgery, and the MPTP-injected NHPs studied post-surgically, as described in Appendix A. FA decreased in MPTP-treated NHPs compared to the healthy group, showing reductions of 6.6% in the right putamen and 5.4% in the left putamen (Appendix A). The MD comparison between healthy and MPTP-treated NHPs showed an increase of 11.4% in the right putamen and 5.8% in the left (Appendix A). It is important to remark that these changes in a t-test do not meet the threshold for statistical confidence, regarding the decreased FA and the increased MD after MPTP administration.

We then compared FA and MD measured after MPTP treatment with the post-operative (POp) condition (Sham or Grafted) after 6 months. The striatum is a heterogeneous structure that includes anatomic/functional subdivisions and several models have been proposed [26,27]. We decided to use the anatomy-functional subdivisions that designate the anterior or limbic putamen, involved in motivation, the medial or associative putamen related to cognition, and the sensorimotor or posterior putamen, linked to locomotion [28], and defined these three regions for further analysis at 6 months post-surgery (Figure 3A).

In the Sham NHP, FA measures decreased after sham surgery in the whole putamina. In sharp contrast, FA increased bilaterally in Grafted 1 and Grafted 2 animals, compared to their previous MPTP condition; Sham showed a decrease and Grafted NHPs showed an increase post-operatively in FA, illustrated by the black lines (Figure 3B). On the other hand, MD values in Sham NHP had discrete changes after surgery, suggesting that there was no effect. Interestingly, Grafted 1 increased in all the analyzed regions, and Grafted 2 showed a consistent bilateral reduction post-grafting (Figure 3C).

A Pearson correlation analysis was performed to correlate behavioral changes with DTI measures in all NHPs in MPTP and POp conditions (Figure 3D). The resulting matrix showed a positive and significant correlation between reaching with displacement and ingestion (0.86 to 0.99). Also, measures of FA in both hemispheres presented a positive correlation of 0.59. For MD, the correlation between the left and right sides was positive and highly significant (0.94). When comparing behavioral tests with FA, values were between −0.72 and −0.38, showing a negative correlation with displacement, reaching, and ingestion, which indicated increased values of FA and decreased times to perform motor tasks.

In contrast, the correlation of both putamina MD values with displacement and reaching were close to zero and reached a positive non-significant correlation with ingestion. As expected, the correlation between FA and MD was negative, with values from −0.66 to −0.52. Thus, behavioral tests showed a consistent negative correlation with FA and found a negative relationship between FA and MD.

### 2.5. PET Qualitative Analysis

We performed 11C-DTBZ PET scanning to assess the presence of DA nerve terminals due to an incomplete lesion or to detect somata of grafted DANs in the putamen. The 11C-DTBZ PPOR was lower in Sham than in Control, and its binding in Grafted 1 and Grafted 2 was greater than in Sham (Figure 4). These data revealed that Grafted NHPs, at 9 months post-surgery, had increased 11C-DTBZ binding in both putamina, indicative of functional DANs at the transplanted site, similar to changes in clinical trials using DANs from induced pluripotent stem cells [18].

### 2.6. Transplanted Neurons Release Dopamine

Before euthanasia, microdialysis was used to quantify the extracellular DA released in the putamen after chemical stimulation in vivo. Probes were inserted simultaneously into both hemispheres of anesthetized NHPs at 10 months POp. The Sham NHP had low basal DA levels and did not present a clear response after high potassium or amphetamine stimulation was administered through the dialysis membrane. In both Grafted NHPs, baseline DA concentrations were elevated compared to Sham. Notably, extracellular DA concentrations sharply increased on both transplanted sides after both stimuli (Figure 5, upper panels, Grafted 1 and 2). The metabolite DOPAC decreased when administering the solution with high potassium and amphetamine in Grafted NHPs (Figure 5 bottom panels, Grafted 1 and 2), consistent with previous data [29]. These results demonstrate that DA was released in both grafted NHPs.

### 2.7. Dopamine Neurons Survive for Ten Months in the Putamen of MPTP-Treated NHPs

After fixation, the brain slices were analyzed for grafted GFP+ neurons in each putamen (Figure 6A), confirming that each individual’s stereotactic coordinates obtained after an MRI were correct. These DANs were TH+, GIRK2+, and MAP2+ (Figure 6B,D). Notably, the grafted animals did not develop tumors and apparently did not elicit an innate or adaptive immune response (Figure 6E), since hematoxylin and eosin staining have been used to identify lymphocytic aggregates [30], or reactive microglia [31], which were not observed in any of the animals. TH counting showed 2.5 × 10^5^ and 2 × 10^5^ cells per hemisphere in Grafted 1 and 2, respectively (Figure 6C). Of note, MPTP injection did not cause a complete loss of DAN cells in the substantia nigra since all animals had 5 × 10^4^ TH+ cells per hemisphere (Figure 6C). It should be noted that glial cells, GABAergic, and serotonergic neurons were also found at lower numbers than DANs in the transplant sites (Appendix A). The low proportion of surviving serotoninergic neurons is consistent with the fact that no dyskinesias were observed in the grafted NHPs. 

## 3. Discussion

Human DANs transplanted in the putamen of parkinsonian NHPs improved some of the analyzed motor behaviors, increased 11C-DTBZ binding, and survived for 10 months without tumors. Importantly, grafted human neurons released DA in the brain after depolarization and amphetamine application through the microdialysis probe, and generated changes in FA and MD, which can be correlated with behavioral performance. However, caution is appropriate since there was a different response of each NHP to intoxication with MPTP, and the number of grafted animals is limited in this work. 

The derivation of midbrain DANs from hESCs was efficient due to the neural-inducing properties of specific small molecules and supported with 76–86% of DANs that co-expressed FOXA2/βIII-tubulin and TH/βIII-tubulin at day 42 of differentiation. Other data that corroborated the identity of these neurons were obtained by RNA-Seq and RT-qPCR, showing the expression of relevant specific markers (*FOXA1*, *FOXA2*, *TH*, *LMX1A*, *SYT4*, *DDC*, *PAX6*, *DCX*, *NEUROG2*, and *ASCL1*). Additionally, *CORIN*, an early DAN progenitor marker, *EN1*, *CNPY1*, *PAX8*, and *HOXA2*, markers for rostrocaudal ventral midbrain patterning are predictive of successful graft outcome [32] and increased during protocol differentiation [20]. Signs of electrophysiological maturation of these neurons were observed at 60 days of culture, including an adaptation in depolarization-induced and spontaneous action potentials; however, other properties of DANs have not been fully developed at this time, specifically K^+^ currents involved in repolarization and the absence of sag. However, we have unpublished indirect evidence that neurons kept maturing when placed in an appropriate environment; action potential duration became briefer, and sag was observed after placing DANs onto rat organotypic cultures (Urrieta-Chávez, in preparation). Transplantation of DANs in the brain of NHPs might favor this maturation process. Furthermore, we observed the release of DA into the medium, an additional characteristic of mature neurons, as described in previous reports [12,15,33,34]. 

Previous reports of grafting DANs differentiated from human stem cells showed that cell therapy induces improvements in parkinsonian NHPs regarding motor behavior and DA uptake sites assessed by PET [16,17,35]. It remains difficult to have a behavioral test with low variability [36], so we decided to associate the motor improvement over a period of 10 months with DA release and imaging analyses. Although the Sham NHP had some months in which the displacement and ingestion were significantly lower than MPTP, this might be due to the plasticity of the MPTP-treated brain [37]. Additionally, in this animal, reaching was more consistent, presenting times that were higher than the basal condition, before MPTP injection. The behavioral effect of MPTP on the Sham NHP was subtle, like that in Grafted 2; the motor changes of Grafted 1 after MPTP were more pronounced. The motor recovery of Grafted 1 was noticeably clear and stable, whereas Grafted 2 presented a more variable response after grafting, but there was an overall significant difference with Sham when the decreased times were compared to the POp months, in relation to the MPTP condition (Figure 2D). The Sham animal showed decreased times to perform the reaching and ingestion tests, an effect that can be related to the use of cyclosporine, as previously reported [38].

Nonetheless, the recovery observed in both Grafted NHPs was correlated with changes in imaging (FA and PET) and in situ DA release. Even though the number of studied animals here was low, we found changes in grafted NHPs with the different approaches. Remarkably, in motor behavioral assessment, graft-induced dyskinesias were not observed, which are related to serotonergic neurons of the graft [39]; their absence may be due to the high percentage of dopaminergic neurons, at least counted in vitro. However, it is essential to note that some authors point out that dyskinesias are probably due to an inflammatory response [40].

The anisotropy of water diffusion in brain tissue was modified by development or disease and was evaluated by FA and MD derived from DTI. In PD, there is a reduction of FA in basal ganglia nuclei [41], like our observations in the putamen of all NHPs after MPTP administration, suggesting microstructural damage to tissue integrity [42,43,44] reflected by FA decrease. MD increases are also related to tissue loss and edema, which we also detected after MPTP administration. The decreased FA and the increased MD might be due to the loss of dopaminergic terminals into the putamen nuclei and the retraction of striatal medium spiny neurons after MPTP denervation [45,46,47]. Interestingly, these changes in FA and MD have been reported for PD patients [48].

For the FA and MD studies, it is worth noting that the values were slightly different between the Sham and Grafted NHPs, with changes measured and compared to the starting values for each NHP. An effect of cell transplantation in both grafted NHPs was suggested by higher FA values in our longitudinal comparison along the putamen. Interestingly, FA values gradually increased, and MD values gradually decreased due to the axonal growth, ion flux across axons, gliosis, and glial scar formation after spinal cord injury in rats [49,50]. Although both Grafted NHPs showed an increase of FA in our study, Grafted 1 showed an increment of MD, whereas Grafted 2 showed a reduction of MD. The increase in FA after grafting can be due to the neuronal processes developed by the transplanted neurons. The fact that Grafted 1 and 2 presented changes in opposite directions can be due to changes in cellularity, with MD increases suggesting edema or cell death [48]; however, the number of surviving DANs was similar in both grafted NHPs. Further, the traumatic brain injury, generated by the needle, can generate changes per se in MD [51]. The changes in the opposite direction of MD have been observed in the phase 1 clinical trial for the severe demyelinating condition, Pelizaeus-Merbacher disease. Two patients of similar age received neural stem cell grafts, showing an increase of FA, but MD increased in one person and decreased in the second [52]; the variable direction change in MD remains to be established but might relate to individual variation. Our correlation analysis of DTI results revealed a negative correlation between FA and behavioral performance and between FA and MD. Such changes, especially increased FA, should be useful to monitor the survival and functioning of DANs in vivo with a non-invasive, non-radioactive technique. The negative correlation between FA and behavior is clear, but the variability in the three putaminal regions and/or the small number of NHPs studied here might have precluded its statistical significance. Notably, the findings in FA changes support the use of DTI as a biomarker for the quantitative analysis of neuronal damage in PD in clinical trials, since its reduction is associated with a lower number of NDA in the substantia nigra [53] of these patients and could have efficiency in diagnosis and prognosis.

Microdialysis had not been used to measure DA levels in grafted NHPs directly. Ten months after grafting, and before euthanasia, grafted neurons released DA with specific stimuli in vivo, like findings reported in rodent PD models [29,54]. Both Grafted NHPs showed higher DA levels than Sham, consistent with previous work reporting sustained behavioral recovery when DA is released by optogenetic stimulation of transplanted DANs [55,56]. Although the concentrations of high-potassium-induced DA release, or the accumulation caused by amphetamine, were lower than those found in non-lesioned brains, the kinetics and the correlation of DOPAC changes associated with the stimuli were consistent with the release of DA by the grafted DANs [29]. Our results cannot rule out an indirect effect of the grafted neurons on the remaining dopaminergic innervation or host-mediated recovery of the endogenous dopamine system. Further studies must establish whether postsynaptic DA receptors do not present supersensitivity after grafting NHPs, a phenomenon reported after DAN transplantation in hemiparkinsonian rats [29]. 

PET analysis showed that grafted DANs were functional in vivo, suggesting the presence of DANs in the brain [57], which may be associated with favorable changes in behavior. MPTP administration to NHPs reduces the binding of dopaminergic PET probes [36,58]. In line with our results, DAN grafting to parkinsonian NHPs consistently increased PET signals, which has been correlated to behavioral improvement [16,59]. Although the acquisition of PET images before MPTP, after injection of this neurotoxin, and the following grafting is ideal for follow-up, we could not have PET scans pre-lesion and after MPTP administration; however, the acquisition of PET with dopaminergic probes is routinely made in presumptive PD patients. Interestingly, 11C-DTBZ uptake in Sham was like the grade 2 “egg shape” pattern reported for PD patients, consisting of bilateral loss of the putamina tracer, with almost normal uptake in the caudate in PET images [60]. In both grafted animals, the pattern of tracer uptake was enhanced in the medial and posterior regions, which is consistent with transplantation sites. These results suggest that the tracer uptake distribution might be important when analyzing grafted and sham primates, in addition to the binding. Another interesting aspect is that Grafted 2 showed variable behavioral recovery and lower PET signal, compared to the stable improvement and higher PET binding observed in Grafted 1. Even though the number of experimental subjects was low in our study, qualitative differences were found between sham and grafted animals, although, no statistically significant differences were found between Grafted and Sham NHP. 

A recent clinical trial used imaging and behavioral assessments as biomarkers after the grafting of NDA and showed poor survival and modest clinical recovery [18]. The microdialysis assay could be an approach but has a great risk of infection as a result of the procedure. However, this release assay could serve as a biomarker for future pre-clinical studies, as it provides additional data on the regulation of DA levels in the grafted brain.

A concern about hESC-based therapies is the development of tumors, considering that few pluripotent cells can remain undifferentiated [61]. Ten months after cell transplantation, NHPs did not show any tumors. Additionally, RNA-seq showed that the pluripotency genes, as well as the cell proliferation gene TTK, decreased upon differentiation, which might partially explain the lack of tumors in Grafted NHPs [62]. Three percent of the total grafted cells were found long-term as DANs. We grafted more cells by hemisphere compared to other studies that employed NHPs (2–3.75 × 10^6^) [12,16,17] or rodent models (3–5 × 10^5^) [11,12,29]. In these reports, 3 ± 1.4% of DANs survived, similar to our results. Despite this, surviving DANs could form new tracts and generate behavioral improvements. These neurons still expressed TH and partially compensated for the loss of DA. According to the development of human pluripotent stem cell-derived DAN replacement therapy in PD, the survival of a minimal number of grafted DANs and enough reinnervation in the putamen are necessary to improve symptoms [13]. These data suggest that motor function improves when a proper number of grafted DANs survive and are functional. 

## 4. Conclusions

The findings of this study show that DANs survive in the putamen and this is associated with a significant improvement in some behavioral tasks, compared with data in the same animals before transplantation, and with DA release in the brain. The imaging studies are suggestive of positive changes, although we did not observe consistent significance differences between the Sham and the Grafted NHP. Importantly, to confirm the beneficial structural and behavioral changes following DAN transplantation, more studies with larger groups of NHPs need to be conducted. 

## 5. Methods

### 5.1. Dopaminergic Differentiation of hESCs

A human embryonic stem cell line H9(WA09)-GFP, which constitutively expressed enhanced Green Fluorescent Protein (EGPF), was used for all experiments [19]. Before differentiation, cells were expanded in a supplemented Knock Out (Gibco^®^, Eugene, OR, USA) medium, conditioned by mitotically inactivated mouse embryonic fibroblasts (MEFs), with fresh FGF-2 (10 ng/mL; Sigma-Aldrich, Saint-Louis, MO, USA), over a Matrigel (BD^®^, Billerica, MA, USA) matrix until they reached 75% confluency. The floor-plate dopaminergic differentiation protocol [12] was performed with minor modifications [9]. Pluripotent hESCs were incubated with small molecules to start the dual SMAD inhibition. BMP and TGF-b receptors were pharmacologically blocked with 100 nM of LDN193189 (Stemgent, Beltsville, USA) and 10 mM of SB431542 (Tocris, Bristol, UK), respectively. The Wnt canonical pathway was stimulated by inhibiting the kinase GSK-3b with 3 mM of CHIR99021 (Stemgent). Shh signaling was stimulated by 1 mM of SAG (Sigma) and 2 mM of Purmorphamine (Stemgent). Recombinant human FGF-8 was added at 100 ng/mL (Peprotech, Rocky Hill, NJ, USA) until day 7. On day 14, cells were cultured in a Neurobasal medium with a B27 supplement, and the morphological changes were evident; cells were positive in a high proportion to NESTIN, indicating that neural progenitors were present at this stage. Neuronal differentiation and survival were promoted by 20 ng/mL of BDNF (Peprotech), 0.2 mM of ascorbic acid (Sigma-Aldrich), 20 ng/mL of GDNF (Peprotech), 1 ng/mL of TGF-β3 (Peprotech), 0.5 mM of dibutyryl cAMP (Sigma-Aldrich), and the Notch inhibitor DAPT at 10 mM (Sigma-Aldrich) in the culture medium (NMM). On day 22, cells were dissociated using TrypLE Express (Life Technologies, Carlsbad, California, USA) and either seeded onto poly-l-ornithine, Fibronectin- and Laminin-treated plates or used for grafting. For electrophysiological and high-performance liquid chromatography (HPLC) assays, cultures were grown over 50 days of differentiation in NMM.

### 5.2. RT-qPCR

RNA was isolated using TRIzol Reagent (Thermo Fisher Scientific, Waltham, MA, USA). Complementary DNA (cDNA) was synthesized by SuperScript III Reverse Transcriptase (Thermo Fisher Scientific) and used for RT-PCR amplification (Taq DNA Polymerase, Thermo Fisher Scientific). Amplification of 50 ng of cDNA was performed with the QuantiFast SYBR Green PCR Master Mix (Qiagen, Germantown, MD, USA) with a StepOnePlus RealTime PCR System (Applied Biosystems, Waltham, MA, USA) with the following primers (Table 1).

### 5.3. Immunocytochemistry

Cells were fixed with 4% paraformaldehyde (PFA), permeabilized and blocked with 0.3% TritonX-100 and 10% normal goat serum in PBS, and incubated overnight with the primary antibodies in PBS plus 10% normal serum (Table 2).

Negative controls without primary antibodies were included and showed no unspecific staining. A Nikon Eclipse TE2000-U microscope was used for image acquisition and the Image-Pro Plus program Version 4.5 (MediaCybernetics, Bethesda, MD, USA) was employed for epifluorescence image analysis.

### 5.4. Karyotype Analysis

Chromosomal analysis was performed by GTG-banding analysis at Reproducción y Genética AGN, Hospital Ángeles del Pedregal, México. Briefly, cells were incubated with colcemid (2 h), harvested by trypsinization, processed with hypotonic solution, and fixed with methanol: acetic acid (3:1). Metaphases were spread on slides, and chromosomes were counted and classified using the G banding technique.

### 5.5. Teratoma Formation Assay

All mouse procedures were performed in accordance with current Mexican legislation (NOM-062-ZOO-1999, SAGARPA, Ciudad de México, Mexico), the Guide for the Care and Use of Laboratory Animals of the National Institutes of Health (NIH) and approved by the Institutional Animal Care and Use Committee of IFC-UNAM. The hESCs were grown on Matrigel (BD Biosciences, Franklin Lakes, NJ, USA), harvested by trypsinization, washed in PBS, and re-suspended in KSR medium with 30% of Matrigel. Cells from a T-25 flask at 80% confluence containing approximately 3–5 × 10^6^ were subcutaneously inoculated into the dorsal flank of each 6–8-week-old nude (nu/nu) mouse. For the presence of cells from the three embryonic germ layers, euthanasia was performed (sodium thiopental, 21 mg/kg, i.p.) and mice were infused intracardially with PFA when tumors reached 4–5 mm. Teratomas were dissected and 20 µm slices were obtained, some were stained with hematoxylin and eosin and other adjacent slices were observed under the fluorescence microscope.

### 5.6. RNA-Seq

Massive next-generation sequencing was performed on MiSeq equipment from Illumina by paired-end reads (2 × 75) using TruSeq RNA Stranded mRNA Library Prep following the provider protocol. About 50.4 million pass-filter reads were obtained. On average, 10.98% reads per sample were obtained (SD = 1.95%). Of these, 3 replicates represented day 0 (11.81%, 13.30%, and 11.62%), four represented day 14 (10.48%, 11.24%, 10.99%, and 12.65%), and two represented day 28 (6.45% and 10.30%) of differentiation. We found similar gene expression patterns when comparing our results with the gene expression levels previously reported by microarrays [12]. The standardized Z-Score was used to illustrate the expression patterns. 

### 5.7. Gene Expression Analysis

We removed adaptor sequences using cutadapt [63] and trimmomatic [64] and obtained 36–76 bp paired-end reads for each sample. The quality of raw sequenced reads was verified using FASTQC [65] (Table 3). Subsequently, we used a previously reported pipeline for read mapping, transcript assembly, and expression estimation [66], and mapped sequenced reads to the human reference genome hg38 (https://www.gencodegenes.org/, accessed on 10 November 2021) by using TopHat v2.1.1 [67] with default parameters. The reads were assembled and mapped using Cufflinks v2.2.1 [68] and calculated the values of Fragments Per Kilobase of transcript per million mapped reads (FPKM) for all annotated genes and transcripts.

The gene annotation was obtained from GENCODE v29 (https://www.gencodegenes.org/, accessed on 10 November 2021) and used HTSeq [69] to calculate read counts for annotated genes. Next, we performed a pairwise comparison of read counts implementing DESeq2 [70]. Genes were labeled as differentially expressed (DEG) if at least one of the replicates in the comparison had FPKM ≥ 1, and normalized count FC > 4 with an FDR < 0.05.

We obtained temporal gene expression profiles of DEGs using hierarchical clustering. In this unsupervised clustering method, we implemented Ward’s linkage algorithm using the Euclidean distance matrix of log2-transformed FPKM values of DEGs.

### 5.8. Electrophysiological Analysis

Whole-cell patch-clamp recordings were performed on 50- to 65-day-old cultures of hESCs differentiated into DANs. Neurons were transferred to a recording chamber and continuously superperfused with an oxygenated saline solution (4–5 mL/min) at room temperature (~25 °C). Micropipettes were pulled (Sutter Instrument, Novato, CA) from borosilicate glass tubes (WPI, Sarasota, FL, USA) to an outer diameter of 1.5 mm for a final DC resistance of 4–6 MΩ when filled with internal saline containing (in mM): 120 KSO_3_CH_4_, 10 NaCl, 10 EGTA, 10 HEPES, 0.5 CaCl_2_, 2 MgCl_2_, 2 ATP-Mg, and 0.3 GTP-Na (pH = 7.3, 290 mOsm/L). Neurons were visualized with infrared differential interference video microscopy and epifluorescent illumination with a 40X immersion objective (0.8 NA; Nikon Instruments, Melville, NY, USA) and a CCD camera (Cool Snap ES2, Photometrics, Tucson, AZ, USA). Recordings were made with an Axopatch 200A amplifier (Axon Instruments, Foster City, CA, USA) and data were acquired with Im-Patch©, CODE VERSION 1.0, open-source software designed in the LabView environment (National Instruments, Mexico City, Mexico; available at www.im-patch.com, accessed on 17 June 2021. The giga-seal resistances were in the range of 10–20 GΩ. The current signals from the amplifier were filtered at 5 kHz through a four-pole low-pass filter.

### 5.9. Lesion of Non-Human Primates with MPTP

Experiments were performed on three adult Vervet monkeys (*Chlorocebus aethiops*), 2 females and 1 male, aged 17–21 years and weighing 4.2–4.5 kg. NHPs were housed in individual cages with 12 h light/dark cycle, room temperature at 23 ± 1 °C, 50 ± 10% of relative humidity, and were fed twice daily with a diet of High protein monkeys LabDiet 5038 biscuits (High Protein Monkey Chow of Lab Chows, Purina^®^ (Ciudad de México, Mexico) water ad libitum, and fresh fruits and vegetables. All procedures were performed in accordance with current Mexican legislation NOM-062-ZOO-1999 (SAGARPA), the Guide for the Care and Use of Laboratory Animals of the National Institutes of Health (NIH) and approved by the Animal Care and Use Committees of Instituto Nacional de Neurología y Neurocirugía (20/13) and Universidad Nacional Autónoma de México. An additional group of three healthy adult NHPs, 2 females and 1 male, with similar weights, were imaged with MRI to study the effect of MPTP. 

Bilateral Parkinsonism was induced in the three NHPs by intramuscular MPTP hydrochloride (Sigma-Aldrich, St. Louis, MO, USA) administration, dissolved in saline at 0.5 mg/kg to reach 2–2.5 mg/kg divided into 4 or 5 daily injections until they developed a clinically evaluated extra-pyramidal syndrome [4,5]. Only one round of injections was needed. MPTP administration was made without anesthesia in a designated special room with the necessary safety measures for the animals and the personnel, following security protocols [71]. After MPTP administration, acute responses included disorientation, mydriasis, and hypersalivation. The developed signs included rigidity and bradykinesia. NHPs were closely monitored by a veterinarian and were provided with water, food pellets, and fresh fruits to maintain their corporal weight and general well-being.

### 5.10. Motor Behavioral Assessment

Parkinsonian motor symptoms were rated using the HALLWAY task previously reported to evaluate the free movements of NHPs [4,5]. Sessions were video recorded daily for five consecutive days. For each monkey, two observers blind to treatment scored motor performance independently. An evaluation session consists of placing two rewards on the lowest shelf just in front of the two holes in the acrylic wall at the end of the hallway; the NHP had to walk through the entire hall to take the reward from this shelf. The NHP was conditioned to return to the beginning of the hallway, and then two additional rewards were placed on the second shelf, repeated for the third and fourth shelves. The second round of the task consists of placing the rewards displaced to the middle of the shelf, forcing the monkeys to use each hand independently to take the reward. In the first round, the primate must return to the beginning of the hallway after taking the rewards from the lowest shelf to place the second, third, and fourth shelves rewards, respectively. Finally, the third round consists of placing four more rewards, one by one, in a well located on the right side of the second shelf to force primates to use a fine grip to take the rewards. Therefore, NHPs had to cross 12 times to take 20 rewards for a successful task. After training, the baseline was recorded for 5 days once the subjects performed more than 95% of the task in less than 10 min. After MPTP intoxication, NHPs had a one-month parkinsonism stabilization period since it is reported that after one month, there is no behavioral improvement even in extended periods, up to 2 years [3,14,16,35,59]. Then, we evaluated the behavioral task for five days. From this point onwards, each session of the hallway task ends when the primate takes all the rewards or after 10 min. After sham or grafting surgery, the animals were evaluated for 5 days monthly, over 300 days (Appendix A). The analysis of the videos consisted of frame-by-frame quantifications of three motor behavioral parameters: (1) Displacement: time to cross the second third of the hallway (1.15 m), (2) Reaching: time to take the rewards, and (3) Ingestion: time to bring the reward to the mouth. 

### 5.11. Magnetic Resonance Imaging

MRI images were acquired in a GE Discovery MR750 (General Electric Healthcare, Milwaukee, WI, USA) scanner of 3T and a commercial 32-channel head coil array at the Instituto de Neurobiología of Universidad Nacional Autónoma de México. The animals were anesthetized with Tiletamine/Zolazepam (2 mg/kg, i.m.) and fixed in a supine position. T1-weighted images were acquired by Fast Spoiled Gradient Echo sequence using the following parameters: TR/TE = 8.5/3.2 ms; FOV = 256 × 256 mm^2^; reconstruction matrix = 256 × 256; final resolution = 1 × 1 × 0.5 mm^3^. The diffusion-weighted imaging (DWI) was acquired by Single Shot Echo Planar Imaging sequence with the following values: TR/TE = 6500/99 ms; FOV = 256 × 256 mm^2^; reconstruction matrix = 256 × 256; final resolution = 1 × 1 × 2 mm; 35 slices; slice thickness = 2 mm; volumes = 65; no diffusion sensitization images with b = 0 s/mm^2^ and 60 DWI of independent directions with b = 2000 s/mm^2^. 

### 5.12. Surgical Procedure and DAN Transplantation

The surgical procedure was performed 6 months after MPTP administration. Lesioned NHPs underwent stereotactic surgery. The surgical coordinates were calculated for each subject from T1 images obtained by MRI (Appendix A). The stereotaxic zero was placed in the middle line at auditory canal levels and we calculated the distance to the target area. Animals were classified by age since sex does not alter this model [3,4] and two monkeys received DAN, designated Grafted 1 (female, 21 years, 4.5 kg) and Grafted 2 (male, 17 years, 4.3 kg), and one monkey received culture medium (Sham; female, 18 years, 4.2 kg). All animals were fasted overnight before surgery and grafted the same day. The induction of anesthesia was performed with Tiletamine/Zolazepam as above. Then, NHPs were intubated and anesthetized with isoflurane (1–2%, O_2_ flow rate of 2 L/min) to maintain a proper anesthetic state. The NHP’s body temperature was maintained by a heating blanket and their head was placed onto the stereotaxic frame in a prone position. Target areas were shaved, and sterilized, and bilateral incisions were made on the scalp to expose the cranial surface. The skull was drilled, making vertical holes to inject the cells into each putamen. Cultures of 22 days were dissociated using TrypLE Express, cell viability was evaluated with trypan blue, and cells were re-suspended at 8.8 × 10^5^ cells/µL. Grafts of DANs were distributed in nine deposits at three different anteroposterior sites totaling 8 × 10^6^ cells per hemisphere. Cells were delivered slowly to three locations on the dorsoventral axis at each injection site. After injection, the burr-hole was sealed with bone wax, and the fascia, muscle, and skin were sutured. Post-operative care started at the end of the surgery and was followed by 1-week with an analgesic (tramadol: 1.4 mg/kg, i.m.) and antibiotic (cephalexin: 25 mg/kg orally). All NHPs received immunosuppression by oral cyclosporine A (15 mg/kg/day; Gel-Pharma, Zapopan, Mexico) treatment which started the day following surgery until euthanasia. The same cell suspension used for grafting was replated on culture plates, fixed, and stained with anti-TH antibodies, confirming the presence of dopamine neurons.

### 5.13. Diffusion Tensor Imaging

DWI scans were preprocessed using the FSL Diffusion Toolbox [72] of FSL 5.0.11 software. Each image was corrected for eddy current distortions and head motion by affine registration to the average b0 image. A binary brain mask was obtained to remove non-brain tissue. The diffusion tensor model was adjusted to generate the FA and MD maps for all NHPs. Each FA and MD map was registered non-linear to the macaque Rhesus template INIA19 [73]. Regions of interest (ROI) of bilateral anterior, medial, and posterior putamen were drawn from the MRI template, and the average of FA and MD for each NHP were computed to the corresponding ROI.

### 5.14. PET Acquisition and Imaging Analysis

PET probes for vesicular monoamine transporter 2 to analyze the functional imaging outcome of cell transplantation were performed 9 months post-surgery. The (+)-α-11C-dihydro tetrabenazine (11C-DTBZ) was synthesized at the Unidad Radiofarmacia-Ciclotrón of Universidad Nacional Autónoma de México in the Tracerlab FXC-Pro synthesis module (GE Healthcare, Uppsala, Sweden) following published procedures [74]. For each animal, initial anesthesia was induced with Tiletamine/Zolazepam (125/125 mg) at 2 mg/kg (i.m.). All animals were fasted overnight before the PET scan. Anesthesia was maintained during the scans with isoflurane (1–2%, O_2_ flow rate of 2 L/min). PET imaging was performed on a Siemens Biograph 64 mCT scanner (Siemens Medical Solutions, Grünwald, Germany) at the Instituto Nacional de Neurología y Neurocirugía. All animals were fixed in the stereotaxic frame in a supine position with the head centered in the FOV and 11C-DTBZ was injected intravenously (74 ± 18.5 Mbq). Transversal PET slices were reconstructed with CT-based attenuation correction using an iterative algorithm (ordered subset expectation maximization (OSEM) + time-of-flight + point spread function), full width at half-maximum (FWHM) kernel of 5 mm, and 5 iterations (21 subsets). Dead time, decay, attenuation, random, and scattering corrections were applied. Spatial normalization by automatic procedures to a common space was applied to reduce intra- and inter-operator variability. First, all images were reoriented parallel to the orbitomeatal axis and unnecessary background data was removed by reducing the matrix size while keeping the voxel size (3D Slicer software, version 5.6.0, www.slicer.org, accessed on 13 January 2021) [75]. Eight 3D T1-weighted MRI images were selected to create a unique template normalized to the Montreal Neurological Institute (MNI) space (McGill, Montreal, QC, Canada), using a rigid 6-parameter registration and normalized correlation as cost function (FSL software) [76,77]. Each PET scan was then registered to a reference 11C-DTBZ non-human primate template previously validated [78] using a non-affine 9-parameter registration and mutual information as to cost function (FSL software). Stereotactically normalized images were smoothed by convolution with an isotropic 3-dimensional Gaussian kernel. Intensity scaling was applied after smoothing. Once a common standard stereotaxic space was established as a reference for the three-dimensional localization, a monkey brain atlas was bilaterally applied to compute the mean intensity within the posterior putamen, cerebellum, and occipital cortex [79,80]. The mean intensity within the cerebellum mask was used as a reference value for scaling, i.e., each voxel value was divided by the reference value. Finally, the posterior-putamen-to-occipital ratio (PPOR) was computed for each subject by dividing the mean intensity within the posterior putamen by the mean intensity within the occipital cortex [81,82].

### 5.15. Microdialysis Experiments and HPLC Analysis

Ten months after grafting, 28-mm long microdialysis probes were manufactured us according to published methods [83]. Animals were anesthetized with Tiletamine/Zolazepam (2 mg/kg, i.m.), maintained by isoflurane through an endotracheal cannula (1–2%, O_2_ flow rate of 2 L/min) and fixed in the stereotaxic frame in a prone position. Using the stereotaxic coordinates, we shaved the target areas, sterilized them, and bilateral incisions were made on the scalp to expose the cranial surface; then, we took out the bone wax to expose the dura. One probe was introduced in each putamen to measure extracellular DA concentrations simultaneously. The active part of the dialysis probe was a polyacrylonitrile membrane (molar weight cutoff, 40 kDa) and its inlet was connected to a syringe mounted on a microperfusion pump. In experiments for in vitro recovery with dialysis, membranes had 15–25% values for DA and 3,4-dihydroxyphenylacetic acid (DOPAC).

Tissue disruption of the probe caused altered neurotransmitter levels, but after 10 min, the values returned to normal, even in human patients [84]. Microdialysis probes were perfused with artificial cerebrospinal fluid at 2 μL per minute for 1 h for tissue stabilization as reported [29,84,85,86] and fractions were collected every 12 min. Monoamines were stabilized by adding 0.1 N of perchloric acid, 0.02% EDTA, and 1% ethanol. Extracellular DA increases were obtained after three basal fractions through bilateral chemical depolarization (isosmotic solution containing 100 mM of KCl, fraction 4) or 30 μM of amphetamine (fraction 8) and quantified for monoamine content by HPLC. No recovery correction was performed. After histological analysis, we found that all microdialysis probes were in the putamina, close to the grafting sites (Figure 6A, left panel).

Twenty µL of dialysate samples were injected into the solvent stream of an HPLC system using a reversed-phase column (C18, 3 µm; 2.1 × 50 mm) coupled to a pre-column (Atlantis, Waters^®^, Milford, MA, USA) with a mobile phase solution containing sodium acetate 25 mM, EDTA 0.01 mM; citric acid 25 mM and 1-octanesulfonic acid 1 mM dissolved in milli-Q water and mixed with acetonitrile in a proportion of 95:5, respectively (pH of 3.35 ± 0.05 at a flow rate of 0.35 mL/min). DA and DOPAC detections were performed by a single-channel electrochemical detector (Waters^®^ model 2465, USA) at 450 mV at a temperature of 30 °C and quantified by peak height measurements against standard solutions. For cultures, samples were collected on day 60 of differentiation [87].

### 5.16. Immunohistochemistry and Postmortem Cell Count

NHPs were perfused via the right carotid with ice-cold saline, followed by PFA. The brain was post-fixed overnight by immersion in PFA and equilibrated in increasing sucrose solutions (10%, 20%, and 30%) at 4 °C. The tissue was sectioned on a cryostat in 20 μm slices and serially recovered on individual slides and some were separated to be stained with hematoxylin and eosin. For immunohistochemical analyses, sections were permeabilized and blocked for 1 h with 0.3% Triton X-100 and 10% normal goat serum in PBS. Samples were incubated overnight at 4 °C with primary antibodies diluted in PBS containing 10% normal goat serum. Alexa-Fluor secondary antibodies were diluted in PBS/10% normal goat serum for 1 h. Nuclei were stained with Hoechst 33258. Immuno-reactive tissue was visualized using an epifluorescence microscope (Nikon Eclipse TE2000-U) or a confocal laser microscope (Carl Zeiss LSM710). Negative controls without primary antibodies were included to confirm the specificity of detection. The number of TH+ in the substantia nigra and TH+/EGFP+ (in both putamen) neurons were quantified in Sham and Grafted NHPs. Twenty-four successive coronal sections throughout the grafts were photographed and the number of cells was calculated at 20X magnification.

### 5.17. Statistical Analysis

For cell cultures and behavioral assessment, we used unifactorial analysis of variance (One-way ANOVA) followed by Tukey’s post hoc analysis and *n* values are from independent experiments (independent differentiations). The DTI results were plotted using the software GraphPad Prism program version 6. Pearson’s correlation matrix between DTI measures and the behavioral outcome (average values from months 1 to 6) was computed in the R 3.3.3 version. The actual *p*-values are in Appendix A and *p* < 0.05 was considered significant.

### 5.18. Single-Case Experimental Design

For each behavior and NHP, the times before the MPTP administration (baseline, B) and the times after the MPTP lesion (M) were measured. As a treatment, the 5 times per corresponding month were considered. Subsequently, the means that resulted from subtracting the MPTP times and the 10-month average times were compared using One-way ANOVA and Tukey’s post hoc analysis [88,89].

## Figures and Tables

**Figure 1 cells-12-02738-f001:**
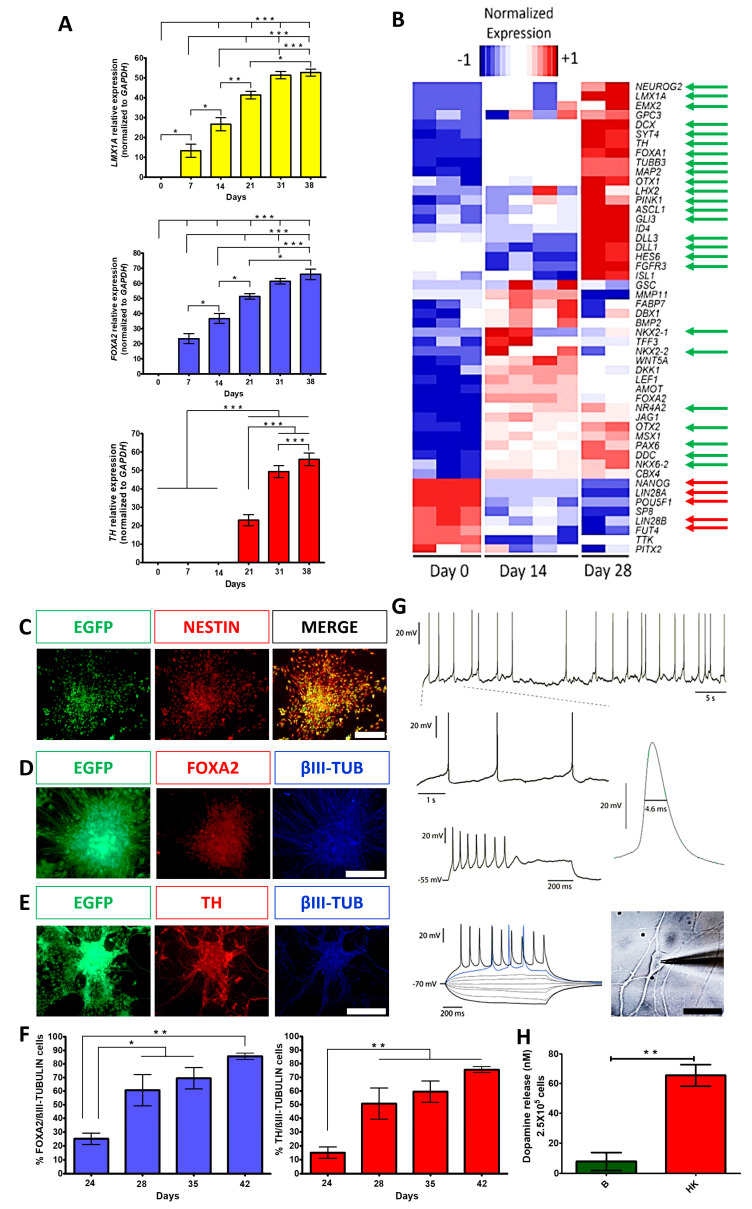
Dopaminergic differentiation of hESCs produced cells with a transcriptional profile and electrophysiological properties characteristic of mature DANs and released DA. (**A**) Gene expression analysis, by RT-qPCR, of dopaminergic markers (*LMX1A*, *FOXA2*, and *TH*) on different days of differentiation, normalized to *GAPDH*. Mean ± SEM; One-way ANOVA followed by Tukey’s test. * *p* < 0.05, ** *p* < 0.01, *** *p* < 0.001, *n* = 5 independent experiments. (**B**) Heatmap from RNA-Seq data showing differential expression of genes at days 0, 14, and 28 of differentiation. Green arrows indicate significant upregulation of relevant DAN markers; red arrows point to significant downregulation of pluripotency markers. (**C**) Immunocytochemistry at day 14 for expression of NESTIN and EGFP. (**D**) Co-expression at day 35 for FOXA2 with βIII-TUBULIN (βIII-TUB). (**E**) Co-expression at day 35 for TH with βIII-TUB. Scale bars for C, B, and D, 200 μm. (**F**) Quantitative co-expression analysis for FOXA2/βIII-TUBULIN (left panel) and TH/βIII-TUBULIN (right panel) at different days of differentiation. Mean ± SEM; One-way ANOVA followed by Tukey’s test. ** p* < 0.05, *** p* < 0.01, *n* = 5 independent experiments. (**G**) Electrophysiological recordings at day 60 show the spontaneous firing of action potentials and membrane potential oscillations at zero currents (−54 mV of membrane potential), characteristic of DA neuron identity (1.5 Hz) (upper panel). Voltage responses to depolarizing and hyperpolarizing current injections (bottom panel) show evoked action potentials and depolarization block (middle panel) at high stimulus intensities. A spontaneous action potential waveform is presented with its duration at half-width indicated; the phase contrast image of a patched neuron is shown on the right panel. Scale bar, 50 μm. (**H**) DA levels, measured by HPLC, in neurons at day 70 of differentiation in basal (B) solution and after chemical depolarization with isosmotic high-potassium (HK) medium. Mean ± SEM; One-way ANOVA followed by Tukey’s test. ** *p* < 0.01, *n* = 5 independent experiments.

**Figure 2 cells-12-02738-f002:**
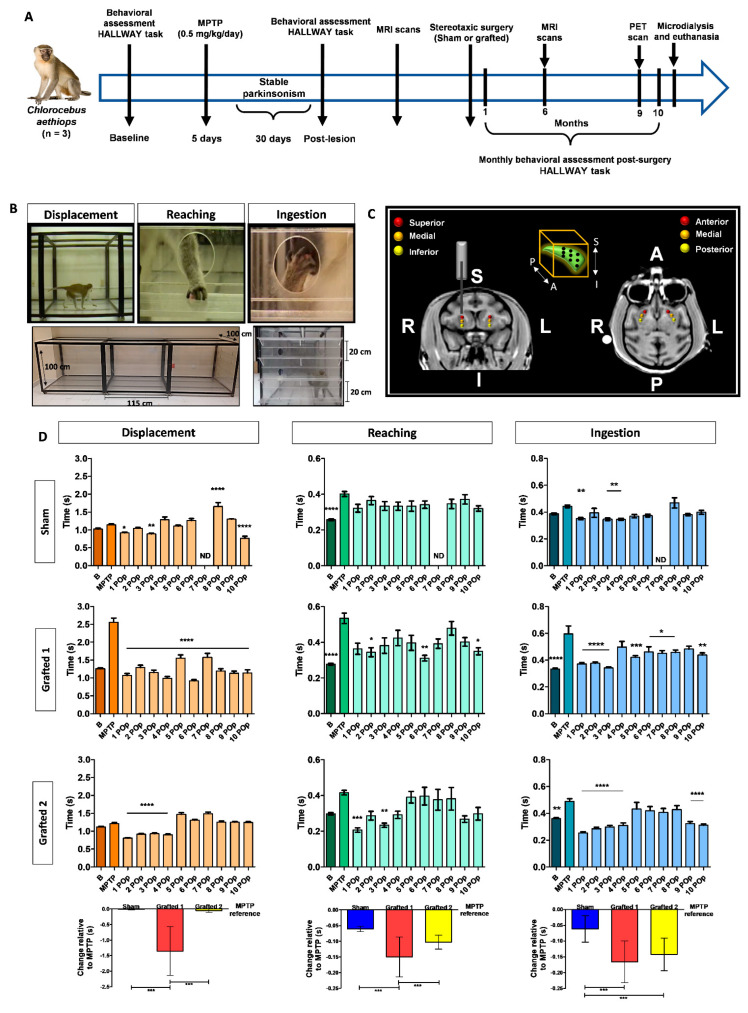
MPTP injection caused behavioral alterations in NHPs that were diminished after DAN transplantation. (**A**) Timeline showing the experimental sequence for NHPs, including behavioral assessment, MPTP administration, DAN grafting, PET, and MRI scans, microdialysis, and euthanasia. (**B**) The panels show the design of the HALLWAY task and representative images of the assessed motor behaviors: displacement, reaching, and ingestion of the reward. The lower images show a lateral and frontal view of the hallway. The animal started the test on the left side of the lateral view. In the frontal image, the 4 shelves are presented. (**C**) MRI images depict the bilateral grafting sites for DANs into the putamen. The 3 injection sites were designated anterior (A), medial, and posterior (P). For these 3 sites, a first injection was made in the inferior (I) part, a second in the medial, and the last in the superior (S) site. The right (R) and left (L) sides are indicated. The box in the middle top part represents the 9 cell deposits in each putamen. (**D**) Graphs show the average time used by each NHP to perform the motor tasks. The bars represent the average times of five consecutive days for each task. Mean ± SEM; One-way ANOVA followed by Tukey’s test. * *p* < 0.05, ** *p* < 0.01, *** *p* < 0.001, **** *p* < 0.0001 relative to the MPTP stage. The bottom part shows the single-case analysis that was used to compare the POp months with MPTP (Appendix A). Mean ± SD; One-way ANOVA followed by Tukey’s test. *** *p* < 0.001. B, basal; MPTP, lesioned; 1–10 Post-operative (POp) month; N.D., Not determined.

**Figure 3 cells-12-02738-f003:**
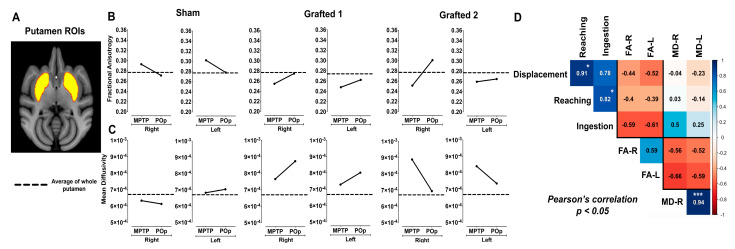
FA and MD measurements in MPTP-treated NHPs after sham surgery or DAN transplantation. (**A**) MRI template and ROI of putamen. (**B**) FA and (**C**) MD quantification in Sham and Grafted NHPs in bilateral putamen in MPTP condition and post-operative (POp), respectively. Reference dotted lines represent the average FA and MD of whole putamen in the healthy group. The black lines represent the average FA and MD of the anterior, medial, and posterior putamen of each subject. (**D**) Pearson’s correlation matrix between the unilateral FA and MD of the whole putamen with the behavioral scores (displacement, reaching, and ingestion) of all NHPs in both MPTP and POp (average values from months 1 to 6) conditions. * *p* < 0.05, *** *p* < 0.001. FA-R, Fractional anisotropy right; FA-L, Fractional anisotropy left; MD-R, Mean diffusivity right; MD-L, Mean diffusivity left.

**Figure 4 cells-12-02738-f004:**
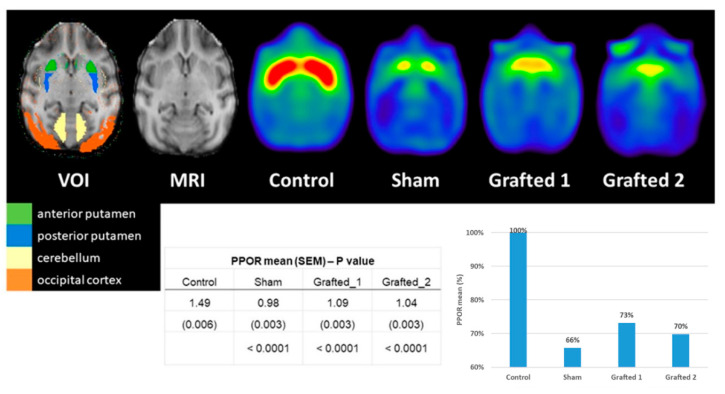
Axial PET sections 9 months after transplantation illustrate a higher binding potential of 11C-DTBZ in the medial and posterior regions of Grafted NHPs. The increased 11C-DTBZ signal was quantified in grafted DAN monkeys at the posterior putamen. The posterior-putamen-to-occipital ratio (PPOR) was higher for Grafted 1 and Grafted 2 than for Sham. VOI, the volume of interest; SEM: standard error of the mean; %: percentage (expressed as a percentage of Control). The control image results from the average of 3 non-lesioned NHPs.

**Figure 5 cells-12-02738-f005:**
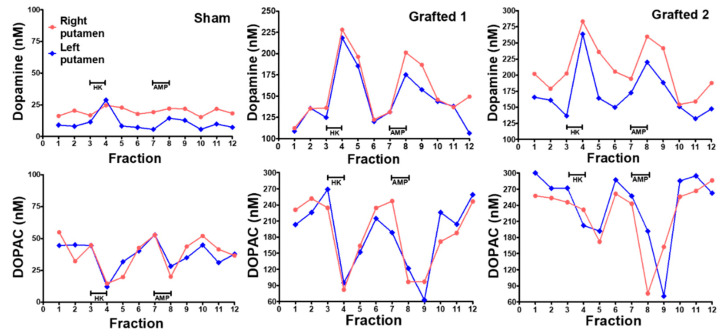
Dopamine was released in the putamen, after chemical stimulation, in Grafted NHPs 10 months post-surgery. Time-course of DA and DOPAC concentrations measured by microdialysis in each putamen after stimulation with 100 mM KCl isosmotic medium (HK) or 30 μM amphetamine (AMP) in Sham, Grafted 1 and Grafted 2 NHPs. Each fraction was collected for 12:30 min. DOPAC, 3,4-dihydroxyphenylacetic acid.

**Figure 6 cells-12-02738-f006:**
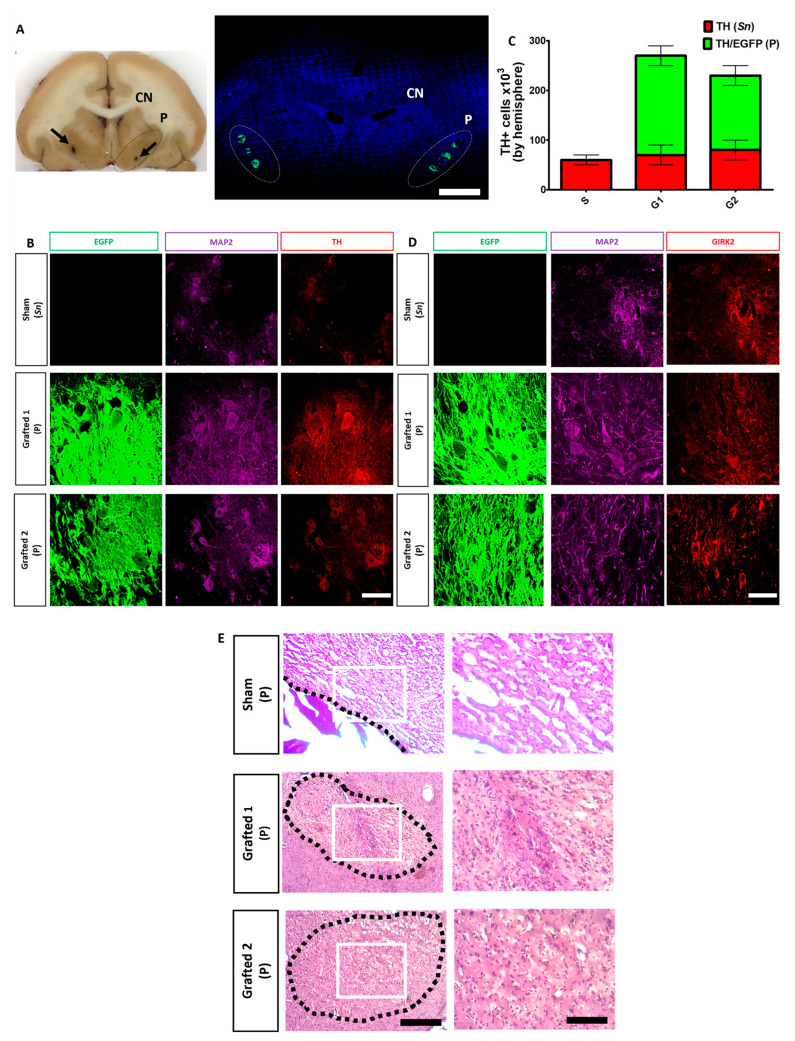
Survival of human DAN grafted in the putamen after 10 months. (**A**) Representative coronal section of the brain (Grafted 1) showing the caudate nucleus (CN) and the putamen (P) (dotted lines). Black arrows indicate the sites of microdialysis probes at each putamen (left panel). Confocal reconstruction of a coronal section is presented on the right panel after nuclear staining (blue) to show the localization of GFP+ transplant sites at each putamen (dotted lines) in Grafted 1. Scale bar, 1 cm. (**B**) Surviving cells in the substantia nigra (Sn) and co-expression of TH (red), MAP2 (cyan), and EGFP (green) by grafted DAN in the putamen (P) after ten months. (**C**) Quantitative analysis for endogenous TH+ (Sn) and TH+/EGFP+ (P) cells after grafting. Mean ± SEM. No significant differences were found. (**D**) Co-expression for GIRK2 (red), MAP2 (cyan), and EGFP (green). (**E**) Hematoxylin and eosin staining at the POp site. Scale bar 200 (left) and 100 μm (right). Dotted lines: site of injection (Sham) or area of grafting (Grafted 1 and 2). White square: area of magnification. Scale bar for B and D, 50 μm. Mean ± SEM. No significant differences were found.

**Table 1 cells-12-02738-t001:** Sequence of primers and targets for RT-qPCR.

Target	Forward (5′–3′)	Reverse (5′–3′)
*LMX1A*	GAGACCACCTGCTTCTACCG	GCCCGCATAACAAACTCATT
*FOXA2*	ATTGCTGGTCGTTTGTTGTG	TGTACGTGTTCATGCCGTTC
*TH*	GAGTACACCGCCGAGGAGATTG	GCGGATATACTGGGTGCACTGG
*SOX2*	TCAGGAGTTGTCAAGGCAGAGAAG	CTCAGTCCTAGTCTTAAAGAGGCAGC
*OCT4*	AGTGAGAGGCAACCTGGAGA	ACACTCGGACCACATCCTTC
*KLF4*	GAACTGACCAGGCACTACCG	TTCTGGCAGTGTGGGTCATA
*GAPDH*	ATGACATCAAGAAGGTGGTG	CATACCAGGAAATGAGCTTG

**Table 2 cells-12-02738-t002:** List of antibodies employed for immunocytochemistry with dilutions, brand, and catalog number.

Antibody	Host	Dilution	Brand	Catalog Number
OCT4	Mouse	1:250	BD Biosciences (Franklin Lakes, NJ, USA)	611202
SOX2	Rabbit	1:500	Abcam (Cambridge, UK)	AB97959
NANOG	Rabbit	1:1000	Peprotech (Rocky Hill, NJ, USA)	500-P236
SSEA4	Mouse	1:400	Abcam	AB16287
NESTIN	Rabbit	1:500	Covance (Daytona Beach, FL, USA)	839801
TH	Rabbit	1:1000 IC 1:500 IH	Pel-Freez Biologicals (Rogers, AR, USA)	P40101
βIII-TUBULIN	Mouse	1:3000	Covance	MMS435P
FOXA2	Rabbit	1:500	Millipore (Burlington, MA, USA)	07633
MAP2	Mouse	1:500	Sigma-Aldrich (Saint-Louis, MO, USA)	M4403
GIRK2	Rabbit	1:1000	Millipore	AB5200
GFAP	Mouse	1:500	Sigma-Aldrich	G3893
GAD 65/67R	Rabbit	1:200	Millipore	AB1511
Serotonin	Rabbit	1:5000	Sigma-Aldrich	S5545
Alexa Fluor 350 Anti-Mouse IgG	Goat	1:1000	Thermo Fisher Scientific	A21049
Alexa Fluor 488 Anti-Mouse IgG	Goat	1:1000	Thermo Fisher Scientific	A11029
Alexa Fluor 568 Anti-Rabbit IgG	Goat	1:1000	Thermo Fisher Scientific	A11036

**Table 3 cells-12-02738-t003:** Detailed information about the total amount of RNA-Seq reads obtained for each sample and the quality controls used to filter the reads. Reads PF: Reads proofread. Trimmomatic was used to remove adaptors from the read sequences. RNA STAR (Spliced Transcripts Alignment to a Reference) allows spliced RNA-seq reads to be mapped into the reference genome. RmDup allows to identify and remove duplicated reads. Day 0, 14, and 28 of the differentiation protocol. r1-r4, replicate number.

Sample	Reads PF	Paired Reads (Trimmomatic)	% Trim	RNA STAR	% STAR	RmDup	RmDup Aprox	RNA STAR Count Unmapped	RNA STAR Count Mapped	RNA STAR Count %	RNA STAR Count ALL %
Day0-r1	2,975,881	2,828,933	95.0%	2,510,778	88.7%	92.5%	2,322,470	557,997	1,952,781	77.78%	65.6%
Day0-r2	3,349,069	3,199,515	95.5%	2,847,575	89.0%	92.3%	2,628,312	637,844	2,209,731	77.60%	65.9%
Day0-r3	2,927,902	2,799,662	95.6%	2,471,487	88.2%	92.6%	2,288,597	552,669	1,918,818	77.64%	65.5%
Day14-r1	2,639,118	2,506,396	94.9%	2,212,330	88.2%	92.5%	2,046,405	474,216	1,738,114	78.56%	65.8%
Day14-r2	2,831,743	2,700,318	95.3%	2,401,428	88.9%	92.4%	2,218,919	498,600	1,902,828	79.24%	67.2%
Day14-r3	2,768,272	2,639,168	95.3%	2,348,184	88.9%	92.5%	2,172,070	497,287	1,850,897	78.82%	66.8%
Day14-r4	3,185,871	3,035,967	95.2%	2,709,484	89.2%	91.7%	2,484,597	569,966	2,139,518	78.96%	67.1%
Day28-r1	1,624,658	1,549,005	95.3%	1,393,858	89.9%	96.1%	1,339,498	255,928	1,137,930	81.64%	70.0%
Day28-r2	2,593,374	2,461,511	94.9%	2,214,291	89.9%	94.7%	2,096,934	412,712	1,801,579	81.36%	69.4%

## Data Availability

The data supporting the findings of this study are available within the article and its Appendix A.

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
