# Peer review of "Human Embryonic Stem Cell-Derived Immature Midbrain Dopaminergic Neurons Transplanted in Parkinsonian Monkeys"

_cells, 2023, doi:10.3390/cells12232738_

Round 1

Reviewer 1 Report (Previous Reviewer 3)

Comments and Suggestions for Authors

The manuscript of López-Ornelas and colleagues encompasses several complementary analyses of aspects of an experimental ES-derived neuron replacement therapy in non-human primates. There are several interesting new findings and, given the use of NHP animal models, this represents an informative exploration of a pre-clinical therapeutic approach. While limited in power, the study is comprehensive in content and does well to expand on studies of similar therapeutic interventions.

The main outstanding issue with the paper is the limited statistical power associated with the use of just three NHPs in combination with the inconsistent outcome of initial MPTP dosing meaning that the effect of DAN engraftment (particularly on behavioral parameters) cannot be shown with a high degree of confidence. However, while manuscript still is somewhat forceful in its claims to the reader, the authors do deal with this limitation and statements are now reasonably in line with what is evidenced in the data. There are a few minor issues which they should address, listed below.

L37:

I presume the authors refer to the data being novel when they say ‘we perform unreported analysis’, but this reads strangely. Suggest replacing with identifying what exactly is not known from existing literature and how your data answers it.

L193:

I think the authors have conflated two of my suggestions here, and the sentence that results does not make sense. I suggested to give either the range parameters from a confidence interval calculation or reference a t-test or non-parametric group-wise analysis, and then regarding this highlight that it does not meet the threshold for statistical confidence. Replace with ‘… do not reach 95% confidence threshold for statistical significance’.

Likewise, the authors should retract the claim of ‘suggesting consistent changes in these parameters’ as it is not sufficiently evidenced. Additionally, please merge the sentences so that the qualification highlighting the lack of significant difference is stated alongside the claimed increase.

L397:

The authors here say that DAN survival correlates with improvement in behavioral tasks, implying a scalar quantitative relationship of some kind. As I understand, the authors did not actually try to relate degree of survival directly with the behavioral tasks, and therefore this is an ‘association’ rather than ‘correlation’

Author Response

Reviewer 2 Report (Previous Reviewer 1)

Comments and Suggestions for Authors

1. Abstract: I strongly recommend that the abstract be modified to include the word "two" as follows "...and transplanted them into two parkinsonian monkeys".  This will avoid misleading readers.

2. Conclusion: I recommend that the end of the first sentence of the conclusion be modified to include the phrase, "compared with data in the same animals before transplantation".

3. Discussion: I view Lines 294/5 as very misleading as they indicate a statistical comparison with Sham, which is not justified.

4. Discussion: Lines 353/4 would be much improved by adding "or host-mediated recovery of the endogenous dopamine system"

5. Discussion: Line 374/5.  in my view this sentence is totally unjustified as it wrongly implies a valid statistical comparison between the one sham and 2 grafted animals.

Round 2

Reviewer 2 Report (Previous Reviewer 1)

Comments and Suggestions for Authors

The balance of the manuscript is improved.  This reviewer recommends a few further edits to improve the readability and improve the accuracy of the data reported.

1. Lines 319-322 are confusing as written.  I suggest the following wording: "For the FA and MD studies, it is worth noting that the values were slightly different between the Sham and Grafted NHPs, with changes measured and compared to the starting values for each NHP. An effect of cell transplantation in both grafted NHPs was suggested by higher FA ….."

2. I recommend that the insert on line 350 ("or host-mediated recovery of the endogenous dopamine system") should be moved to line 358, where it makes sense.  Specifically: Our results cannot rule out an indirect effect of the grafted neurons on the remaining dopaminergic innervation or host-mediated recovery of the endogenous dopamine system".

3. Line 378: Recommend to insert "qualitative" before the word "differences" to clarify meaning here.

4. I noted that lines 404-406 are repeated on lines 406-408.

Author Response

This manuscript is a resubmission of an earlier submission. The following is a list of the peer review reports and author responses from that submission.

Round 1

Reviewer 1 Report

Comments and Suggestions for Authors

This reviewer appreciates that much work has gone into this study on an important topic, but basically this is just a preliminary/case report at this stage. The overall problem is that the inadequacies of the study are not clearly acknowledged (i.e., using a variable PD model, using only n=1/2 animals per group with no between-group statistics) and yet it concludes that the work supports use of this procedure in patients.

The first issue is the well-known variability in the nonhuman primate MPTP model, meaning that to reach a firm conclusion on a hypothesis involving MPTP, a power analysis needs to be conducted beforehand and the derived (large) number of monkeys in each group used in the subsequent study.

The second issue is that the authors use statistics to show changes across time in individual monkeys but do not and cannot meaningfully compare responses between groups (which is the only important comparison), as this reduces the statistics to a group with one subject compared to a group with two subjects.

Author Response

Response to Reviewer 1 Comments

This reviewer appreciates that much work has gone into this study on an important topic, but basically this is just a preliminary/case report at this stage. The overall problem is that the inadequacies of the study are not clearly acknowledged (i.e., using a variable PD model, using only n=1/2 animals per group with no between-group statistics) and yet it concludes that the work supports use of this procedure in patients.

We thank you for this comment. To acknowledge the limitations of the study, we have modified the text to mention those in some sections: the variability in the response of NHPs to MPTP is mentioned in the results and discussion. The low number of animals which is a limitation is mentioned, and the abstract and conclusion sections do not refer to the clinical translation to patients.

The first issue is the well-known variability in the nonhuman primate MPTP model, meaning that to reach a firm conclusion on a hypothesis involving MPTP, a power analysis needs to be conducted beforehand and the derived (large) number of monkeys in each group used in the subsequent study.

We deeply appreciate the comments to the manuscript, and we are aware of the limitation for the n of animals in the groups studied, so we complement behavior with imaging studies and biochemical assays that could add valuable information.

Abstract was modified to include the following sentence:

“Although these results need to be confirmed with larger groups of NHPs, our molecular, behavioral, biochemical, and imaging findings support the integration and survival of human DAN in this pre-clinical PD model”.

Attending to your important comment in the conclusion, the following changes were made:

“The findings of this study support at the molecular, biochemical, imaging, and behavioral levels, the notion that stem cells differentiated to DAN can be grafted into MPTP NHP PD model, where they integrate, survive, and release DA which results in behavioral changes. Importantly, to confirm beneficial structural and behavioral changes following DAN transplantation, more studies with larger groups of NHPs need to be conducted”.

The second issue is that the authors use statistics to show changes across time in individual monkeys but do not and cannot meaningfully compare responses between groups (which is the only important comparison), as this reduces the statistics to a group with one subject compared to a group with two subjects.

Thank you very much for the comment. We agree that the most important statistical analysis would have been between groups, which unfortunately could not be performed given the n. However, the individual analysis was also of great importance and showed that the lesion with MPTP was inconsistent in NHPs. Without this analysis, we would have mistakenly assumed the misconception that the lesion with MPTP was stable in all animals since at first impression the times in all behaviors evaluated after the MPTP lesion increased, although marginally in some cases (MPTP bars in Fig.2D).

Given our limited number of NHPs, we now performed a Single-case experimental design (http://dx.doi.org/10.1037/pri0000029) with Sham and Grafted animals (Supplementary Figure 4) and added the Single-case experimental design section in Methods. In all cases and behaviors, the post-lesion times after MPTP administration increased, with different levels, being higher in Grafted NHPs, for reaching and ingestion behaviors. In displacement behavior, the highest response was in Grafted 1. In all cases, the decrease in times to perform the tasks suggests that grafting dopamine neurons was responsible for the effect.

The effects of grafting compared with lesion with MPTP indicated that in Sham there was a decrease in trend in all behaviors, being significant in reaching and ingestion. This slight effect may be due to the administration of cyclosporine, which inhibits neuroinflammation and has been associated with improvements in the PD model (https://doi.org/10.1038/s41586-023-06300-4). However, the levels of the effect were higher in Grafted NHPs, and short latency indicates that the effects were generated by the NDAs. Comparison of the means between the Grafted and the Sham showed significant differences in reaching and ingestion tasks (Fig. 2D, new graphs added at the bottom).

Reviewer 2 Report

Comments and Suggestions for Authors

The authors have added additional valuable graft characterization data in the revised version of the manuscript, and they have responded adequately to my comments. However, given the many uncertainties of the study (only 1 control and 2 treated animals, incomplete lesioning and spontaneous recovery in the MPTP model), I suggest that the title of the paper is moderated to not draw conclusions on recovery. A suitable title would be to simply remove the last 6 words, leaving the title as “Human Embryonic Stem Cell-derived Immature Midbrain Dopaminergic Neurons Transplanted in Parkinsonian Monkeys”

Minor comment:

There is a typo in Page 7 (“grated”)

Author Response

Response to Reviewer 2 Comments

The authors have added additional valuable graft characterization data in the revised version of the manuscript, and they have responded adequately to my comments. However, given the many uncertainties of the study (only 1 control and 2 treated animals, incomplete lesioning and spontaneous recovery in the MPTP model), I suggest that the title of the paper is moderated to not draw conclusions on recovery. A suitable title would be to simply remove the last 6 words, leaving the title as “Human Embryonic Stem Cell-derived Immature Midbrain Dopaminergic Neurons Transplanted in Parkinsonian Monkeys.”

We thank comments on the new data provided. According to your suggestion, the title of the article was changed.

Minor comment:

There is a typo in Page 7 (“grated”)

Thank you for the observation and the word was corrected.

Reviewer 3 Report

Comments and Suggestions for Authors

The manuscript of López-Ornelas and colleagues encompasses several complementary analyses of aspects of an experimental ES-derived neuron replacement therapy in non-human primates. There are some issues with presentation and figure legend detailing but one appreciates a generally comprehensive and considered display of the data contained. With regard to the behavioural assessment, the authors sometimes stretch beyond what is fair assessment based on the data they present. One appreciates that with only three animal subjects the statistical power in this experiment is low, and lack of strong effects does not preclude the possibility. In any case though, I do not believe that the outcome of the behavioural studies should preclude interest in subsequent imaging and dopaminergic functional studies, so it is better to maintain a more restrained tone in the interpretation of these data.

In the imaging and dopaminergic neuron assessment, there are some very clear results, including a degree of novelty with regard to dopamine concentration and response. The authors, though, could provide some more exploration of how this could shape future pre-clinical or clinical studies. For neurodegenerative conditions target engagement biomarkers are of great importance, and I feel there is more that could be developed in the discussion from this data in particular.

In several points there is a blurring of statistically evidenced changes and mere changes in mean. Relatedly, there is a tendency to over-interpret difference that are most likely just noise in the data. With only 3 animals I think it is essential to focus on the big picture to identify what is truly related to the experimental intervention and not just idiosyncrasies related to individual animal subjects.

I include a line-by-line list of comments and suggestions below. Several of these I would consider critical to address (highlighted bold) and should either be revised or defended with well-justified rebuttal. Once  complete, though, I believe this manuscript should be of good merit and rigor and progress to publication.

L96:

-          The authors refer to ‘midbrain floor-plate DAN’. As the DA neurons are mature they are no long floor-plate precursors, correct? So would be better termed ‘midbrain-floor-plate-derived DANs’ or ‘DANs, derived via midbrain floor-plate precursors, as previously reported []’

L98

-          Please clarify in the initial sentence that the DA neuron markers here are LMX1A and FOX2A

L104

-          The authors should make specific comment on the changes in FOXA2 in the RNAseq data.

L113

-          The authors should clarify that FOXA2 is a ventral-midbrain DAN marker.

L147

-          I suggest the authors give a brief overview of the delivery and dosing of the MPTP NHPs as these are introduced, to save the reader from immediately turning to the methods to establish this.

-          The statement ‘motor alterations more like patient than rodent models’ is imprecise and without evidence or citation. The authors should refine their justification here.

L162

-          The lack of a significant different between ‘MPTP’ and baseline in the ‘displacement’ and ‘ingestion’ tasks is a major barrier to making valid conclusions from this data, as well as the high degree of month-on-month variability following this (particularly for the ‘displacement’ task). Firstly, the MPTP model is known to cause a chronic and irreversible parkinsonian phenotype. To not see this in the sham-treated animal imposes doubt on the reliability of the assay or the model.  The most positive interpretation would be that the individual reliability of the measure in each 5-day period is weak, leaving open the validity of data collected over multiple timepoints. However, we require more evidence and argument from the authors if we are to take this interpretation.

-          Given this, we are liable to question if the change seen in ‘Grafted 1’ is truly a response, or simply an unusually (but not anomalous) recording in the ‘MPTP’ timepoint followed by a regression to mean in the follow timepoints.

-          While the increase in ‘Reaching’ task time following MPTP treatment is more consistent than for ‘Displacement’ and ‘Ingestion’, there is nonetheless still a similar degree of reduction in time to complete in the sham-treated animal as the DAN-treated animals.

-          The decreased in task time at months 1-4 in ‘Grafted 2’ is the most convincing of the data here in terms of evidencing efficacy. However, there are concerns if this itself may be strongly impacted if baseline measurements are inconsistent.

-          I do not necessarily think that these issues invalidate the later interesting contributions which the authors make, but statements on the efficacy of the treatment evidenced by this data should be toned down. I do not think that the statement ‘Grafted 1 showed significantly decreased performance times lasting ten months’ is fair considering this and the authors should highlight the caveats in ‘Grafted 1’.

-          In this section in particular there is a mixed use of Arabic numeral and word-form numbers. This is an issue throughout and the authors should revise with care.

L178

-          No statistical justification is given for the claim of decreased FA and increased MD. The authors say themselves that there is no significant difference. Given this, there is no confidence that there is a difference and the authors are not justified in making this statement unless they can give proper justification for this claim.

L191

-          I do not see it is fair statement to say ‘In the Sham NHP, FA measures decreased after sham surgery, except for the left anterior putamen’. In this case although the overall putamen measurement is decreasing, between the left and right sub-divisions there is an exactly opposite trend of change of Sham FA with respect to the scale of change and relative contribution to the change from each division.

-          Similar goes for the statement on Grafted 2 posterior left side.

-          If we are to accept that reduction versus increase in the FA of the putamen is indeed a replicable finding (this does seem a reasonably evidenced claim) then to accept this as reflective of the real sub-division alterations seems fanciful. Instead, it seems likely that measurement of the sub-division serves mainly to introduce noise to the data, which may be more robust simply keeping together in the ‘whole putamen’. Indeed, in the grafted animals there is little in the way of consistent sub-division-level trends in either DA or MD, suggesting that this analysis adds little and relegation of this element to the supplementary data could aid the clarify of the main figure.

L196

-          I would hesitate to call this ‘a variable response after surgery’ – seems most likely there is no effect and this is within margin of error of the assay, unless there is evidence to suggest otherwise?

L208

-          In the statement ‘right and left putamen MD values were close to zero for displacement’ – this is referring to correlation with displacement (rather than actual value) but it is not worded as such.

L210

-          The authors say ‘Interestingly, the correlation between FA and MD was negative…’ – could the authors clarify why this is interesting? My impression was that this was expected and part of the premise. Which, while reassuring to see, is not ‘interesting’ (in the sense of a finding informative of the biology). Perhaps I am interpreting this wrong, in such a case I think it would help the readership if this could be expanded a little.

L214

-          This section has significant repetition of highly similar statements. As I understand, the finding is (a) 11C-DTBZ PPOR is lower in Sham than control, (b) in Grafted 1 and Grafted 2 11C-DTBZ PPOR is greater than in Sham, indicating increase functional DAN (note below). This can be stated more clearly in a single sentence.

-          The authors say that this is indicative of DAN function. They should provide justification for this, or citation to where we can be assured that this is the case.

L225

-          The authors should add figure references each time they are specifically referring to data in Figure 5.

L236

-          The H&E staining evidencing ‘[apparent lack of] innate or adaptive immune response’ seems a superficial analysis. Could these be staining for microglial, macrophage markers etc to add weight to this statement.

L243

-          Correct spelling of ‘grafted’

L282

-          Revise these summaries in the context of the earlier review comments.

L287

-          ‘..of the same [32]’ is there a missing word here?

L302

-          ‘functional recovery … was a accompanied’ I think you mean ‘was suggest by’ – it is a stretch to say that functional recovery has simply occurred.

L311

-          In Grafted 2 – does it really seem likely that the ‘edema or cell death’ is occurring at any great level, as the authors allude to with this sentence, when considering the all the evidence in the study? Seems an unhelpful suggestion unless the authors feel it is a real possibility that has not been excluded by the histological and other data they later covered.

L313

-          Should give further detail to the effect of ‘…phase 1 clinical trial for the severe congenital demyelinating condition, Pelizaeus-Merbacher disease’.

L364

-          Did the authors mean to say ‘…compensate for the loss of endogenous DA neurons’ or something to this effect here?

L371

-          Could the authors also add something on what their findings might suggest for the use of FA as a biomarker in clinical trials?

Methods:

L438

-          The authors must give more detail on the bioinformatic methods used in the RNAseq analysis, as well as details of the quality control from this such as genome coverage etc.

L524

-          Sex does not alter this model 3,4 and two…’. Are 3 and 4 supposed to be references? They are not formatted as such.

Figures

L873

-          The image quality in figures 1C-E is too low. Could the authors please check that the images they are submitting are of a high standard and indicate if the issue is at their end or between the journal and the reviewers so this can be resolved.

L877

-          In Figure 1A, it is unclear what comparisons the indicated statistical differences refer to. The statistical method used is not indicated in figure legend, and it appears that multiple types of analysis were used. The authors must clarify this.

-          It is unclear what is meant by ‘independent experiments’. Are these independent differentiations? Parallel differentiations split from the same originator population? More detail is needed to assess exactly how independent these are from each other. This comment goes for other figure legends where this phrase is used.

-          With these resolved I think the figure is satisfactory to show the point, but I wonder if normalizing only to GAPDH is robust; as, while stable within a particular line, GAPDH can change considerably in expression between cells of different lineages. The authors should consider using several ‘normal expression’ genes to counter this.

L881

-          The authors should clarify how these genes were selected for display in this figure. Presumably they ae ‘top hits’ of some kind, but there should be detail of where they came and from what analysis. Some of this detail may be better placed in the methods, but it should be somewhere.

L884

-          The authors should detail what is  meant by ‘quantitative co-expression analysis’, with summary of numbers of cells used, replicates etc.

L892

-          Please clarify in the figure legend for figure 1H that DA concentrations were measured by HPLC.

L897

-          In figure 2D, non-standardization of y axis limits within each of the motor task results impairs interpretation, the authors should correct his.

L930

-          I am not experienced with these types of analysis and perhaps it is standard, but ‘percentage reduction (expressed as a percentage of Control)’ is a confusing metric to me. Suggest changing simply to % of control, with sham having 65%, Grafted 1 having 75%, Grafted 2 having 71% etc, or simply showing the PPOR ratios with SEM (as in the table) in a bar chart – I think it communicates the exact information.

Supplement

Supp. Fig 1D:

-          The authors do not detail the histological methods used to obtain these images. They must include this in the methods or at least in the supplementary figure legend. Additionally, the authors should help the reader by clarification on the relationship between the H&E staining in the upper panel and the fluorescence imaging in the lower panel. Were these from the same sample, adjacent sections etc?

Supp. Fig. 4

-          The authors title the figure legend that FA and MD decrease and increase, respectively. But no statistical justification is given for this sentence.

-          The authors should indicate the individual mice within each set, so that it is shown which are on the left side and right side and how these relate. This can be done by changing point shapes accordingly.

Author Response

Response to Reviewer 3 Comments

The manuscript of López-Ornelas and colleagues encompasses several complementary analyses of aspects of an experimental ES-derived neuron replacement therapy in non-human primates. There are some issues with presentation and figure legend detailing but one appreciates a generally comprehensive and considered display of the data contained. With regard to the behavioural assessment, the authors sometimes stretch beyond what is fair assessment based on the data they present. One appreciates that with only three animal subjects the statistical power in this experiment is low, and lack of strong effects does not preclude the possibility. In any case though, I do not believe that the outcome of the behavioural studies should preclude interest in subsequent imaging and dopaminergic functional studies, so it is better to maintain a more restrained tone in the interpretation of these data.

We thank the comment. Regarding the behavioral data, we have acknowledged the variable response in behavioral tests after MPTP injection and mentioned this, together with the low number of animals, as limitations of our study in the manuscript. We also changed the description of results in the behavior section to restrain the tone in the interpretation of the findings. In addition, after performing a single-case statistical analysis, we were able to compare the overall response after MPTP between the sham and grafted animals (Fig. 2D bottom part and Supplementary Figure 4).

In the imaging and dopaminergic neuron assessment, there are some very clear results, including a degree of novelty with regard to dopamine concentration and response. The authors, though, could provide some more exploration of how this could shape future pre-clinical or clinical studies. For neurodegenerative conditions target engagement biomarkers are of great importance, and I feel there is more that could be developed in the discussion from this data in particular.

We thank the observation and now we added the next statement in the Discussion section:

“Nowadays, clinical trials use imaging and behavioral assessments as biomarkers after grafting of NDA and show poor survival and modest clinical recovery [29]. The microdialysis assay could be an approach but has a great risk of infection as a result of the procedure. However, this release assay could serve as a biomarker for future pre-clinical studies, as it provides additional data on the regulation of DA levels in the grafted brain.”.

The suggestion to discuss target engagement biomarkers for PD is difficult to apply in this study, because this term relates to the percentage of a target protein that is bound by a drug.

In several points there is a blurring of statistically evidenced changes and mere changes in mean. Relatedly, there is a tendency to over-interpret difference that are most likely just noise in the data. With only 3 animals I think it is essential to focus on the big picture to identify what is truly related to the experimental intervention and not just idiosyncrasies related to individual animal subjects.

Thank you for the comment. Given our limited number of NHPs, we now performed a Single-case experimental analysis (http://dx.doi.org/10.1037/pri0000029) with Sham and Grafted animals (Supplementary Figure 4), added the Single-case experimental design section in Methods and new graphs were included at the bottom of Fig. 2D, where the decrease in times for each task is shown, comparing the grouped data from months 1-10, with the MPTP values. With this single-case analysis we observed that for all behaviors, times after MPTP increased, at different levels, being higher in Grafted NHPs in reaching and ingestion. In displacement behavior, the higher level of response was in Grafted 1. In all cases, shorter times for the tasks suggested that the treatment was responsible for the effect, which was confirmed in the statistical comparison now presented in Figure 2D. In particular for Ingestion, both grafted NHPs are different from the Sham animal. We incorporated these results in lines 173-183 of the current version.

I include a line-by-line list of comments and suggestions below. Several of these I would consider critical to address (highlighted bold) and should either be revised or defended with well-justified rebuttal. Once  complete, though, I believe this manuscript should be of good merit and rigor and progress to publication.

L96:

-          The authors refer to ‘midbrain floor-plate DAN’. As the DA neurons are mature they are no long floor-plate precursors, correct? So would be better termed ‘midbrain-floor-plate-derived DANs’ or ‘DANs, derived via midbrain floor-plate precursors, as previously reported []’

The comment is appropriate, so we changed this phrase to: “Differentiation to midbrain floor-plate-derived DANs was induced, as reported [12]”

L98

-          Please clarify in the initial sentence that the DA neuron markers here are LMX1A and FOX2A

We modified the sentence as follows:

“We confirmed the expression of relevant ventral midbrain (FOXA2) and dopaminergic (LMX1A) markers in agreement with a previous study [12]”

L104

-          The authors should make specific comment on the changes in FOXA2 in the RNAseq data.

We added the next statement:

“The expression of FOXA2 at day 0 of the differentiation protocol is undetected (0 FPKM) and increases at day 14 (26.5 ± 9 FPKM) to have a decrease at day 28 (6.5 ± 2.5 FPKM) of the protocol, a similar expression pattern is observed in 2D platforms during the transition of mature DAN [20]. FOXA1, another ventral midbrain marker, had an increase through the protocol differentiation, as expected. Both factors are involved in the development, specification, and maturation of the physiological functions of DAN [21].”

Furthermore, immunodetection of FOXA2 can be detected at day 24 and progressively increases until day 42 (Fig. 1F).

L113

-          The authors should clarify that FOXA2 is a ventral-midbrain DAN marker.

We clarified with the next statement: “On day 35, expression of the ventral midbrain DAN marker (FOXA2)…..”

L147

-          I suggest the authors give a brief overview of the delivery and dosing of the MPTP NHPs as these are introduced, to save the reader from immediately turning to the methods to establish this.

We added:

“treated with MPTP administered through multiple intramuscular injections (0.5 mg/kg to reach 2-2.5 mg/kg divided into 4 or 5 daily), which developed motor alterations similar to those seen in patients”.

-          The statement ‘motor alterations more like patient than rodent models’ is imprecise and without evidence or citation. The authors should refine their justification here.

We refined the next statement:

…which developed motor alterations similar to those seen in patients [25].

L162

-          The lack of a significant different between ‘MPTP’ and baseline in the ‘displacement’ and ‘ingestion’ tasks is a major barrier to making valid conclusions from this data, as well as the high degree of month-on-month variability following this (particularly for the ‘displacement’ task). Firstly, the MPTP model is known to cause a chronic and irreversible parkinsonian phenotype. To not see this in the sham-treated animal imposes doubt on the reliability of the assay or the model.  The most positive interpretation would be that the individual reliability of the measure in each 5-day period is weak, leaving open the validity of data collected over multiple timepoints. However, we require more evidence and argument from the authors if we are to take this interpretation.

Although there are not significant differences, in all cases, the times to perform the tasks increased after MPTP administration. The variability in the 5 days for basal, MPTP and POp conditions can now be appreciated in the Supplemental Figure 4. As presented now in the new graphs of Figure 2D, there is indeed a decrease in times in the Sham animal in Reaching and Ingestion, displacement. This effect may be due to the administration of cyclosporine, which inhibits neuroinflammation and has been associated with improvements in the PD model https://doi.org/10.1038/s41586-023-06300-4. This is now mentioned in Discussion (lines 295-297).

-          Given this, we are liable to question if the change seen in ‘Grafted 1’ is truly a response, or simply an unusually (but not anomalous) recording in the ‘MPTP’ timepoint followed by a regression to mean in the follow timepoints.

Grafted 1 had a consistent behavioral recovery, as evidenced by the fact that all POp measurements resulted in lower values than MPTP, in the 3 behavioral tasks (Supplemental Figure 4).

-          While the increase in ‘Reaching’ task time following MPTP treatment is more consistent than for ‘Displacement’ and ‘Ingestion’, there is nonetheless still a similar degree of reduction in time to complete in the sham-treated animal as the DAN-treated animals.

As presented in the new graphs in Figure 2D, the decreases in time after MPTP are only comparable between Sham and Grafted 2 in Displacement. For Reaching, both grafted NHPs showed higher decreases, although Grafted 2 is not significantly different from Sham. However, for Ingestion, Sham had significant lower reductions than Grafted 1 and Grafted 2.

-          The decreased in task time at months 1-4 in ‘Grafted 2’ is the most convincing of the data here in terms of evidencing efficacy. However, there are concerns if this itself may be strongly impacted if baseline measurements are inconsistent.

The new analysis showed consistent basal times for the 3 animals. We now support the notion that Grafted 2 also presented significant decreases in Ingestion in the POp period than Sham.

-          I do not necessarily think that these issues invalidate the later interesting contributions which the authors make, but statements on the efficacy of the treatment evidenced by this data should be toned down. I do not think that the statement ‘Grafted 1 showed significantly decreased performance times lasting ten months’ is fair considering this and the authors should highlight the caveats in ‘Grafted 1’.

We thank all the detailed comments. The new data confirms that Grafted 1 significantly improved in all months POp, in the 3 tasks. Notwithstanding, due to the difficulties of the model and the limited number of animals, we have toned down the conclusions.

-          In this section in particular there is a mixed use of Arabic numeral and word-form numbers. This is an issue throughout and the authors should revise with care.

All numbers are now in Arabic numerals.

L178

-          No statistical justification is given for the claim of decreased FA and increased MD. The authors say themselves that there is no significant difference. Given this, there is no confidence that there is a difference and the authors are not justified in making this statement unless they can give proper justification for this claim.

As correctly pointed, we mentioned that these differences are not significant. In the conclusion section, now is the statement “Importantly, to confirm beneficial structural and behavioral changes following DAN transplantation, more studies with larger groups of NHPs need to be conducted”.

L191

-          I do not see it is fair statement to say ‘In the Sham NHP, FA measures decreased after sham surgery, except for the left anterior putamen’. In this case although the overall putamen measurement is decreasing, between the left and right sub-divisions there is an exactly opposite trend of change of Sham FA with respect to the scale of change and relative contribution to the change from each division.

-          Similar goes for the statement on Grafted 2 posterior left side.

-          If we are to accept that reduction versus increase in the FA of the putamen is indeed a replicable finding (this does seem a reasonably evidenced claim) then to accept this as reflective of the real sub-division alterations seems fanciful. Instead, it seems likely that measurement of the sub-division serves mainly to introduce noise to the data, which may be more robust simply keeping together in the ‘whole putamen’. Indeed, in the grafted animals there is little in the way of consistent sub-division-level trends in either DA or MD, suggesting that this analysis adds little and relegation of this element to the supplementary data could aid the clarify of the main figure.

After the suggestion, we changed Figure 3 to show the effect in the whole putamen, and changed the text as follows (lines 262-264):

“In the Sham NHP, FA measures decreased after sham surgery in the whole putamina. In sharp contrast, FA increased bilaterally in Grafted 1 and Grafted 2 animals, compared to their previous MPTP condition;”

L196

-          I would hesitate to call this ‘a variable response after surgery’ – seems most likely there is no effect and this is within margin of error of the assay, unless there is evidence to suggest otherwise?

The statement now is:

“MD values in Sham NHP had discrete changes after surgery, suggesting that there is no effect”.

L208

-          In the statement ‘right and left putamen MD values were close to zero for displacement’ – this is referring to correlation with displacement (rather than actual value) but it is not worded as such.

This phrase was updated: “the correlation of both putamina MD values with displacement and reaching were close to zero”.

L210

-          The authors say ‘Interestingly, the correlation between FA and MD was negative…’ – could the authors clarify why this is interesting? My impression was that this was expected and part of the premise. Which, while reassuring to see, is not ‘interesting’ (in the sense of a finding informative of the biology). Perhaps I am interpreting this wrong, in such a case I think it would help the readership if this could be expanded a little.

Interestingly was changed to “As expected”.

L214

-          This section has significant repetition of highly similar statements. As I understand, the finding is (a) 11C-DTBZ PPOR is lower in Sham than control, (b) in Grafted 1 and Grafted 2 11C-DTBZ PPOR is greater than in Sham, indicating increase functional DAN (note below). This can be stated more clearly in a single sentence.

-          The authors say that this is indicative of DAN function. They should provide justification for this, or citation to where we can be assured that this is the case.

We modified the manuscript with the next statement:

11C-DTBZ PPOR was lower in Sham than in Control, and its binding in Grafted 1 and Grafted 2 was greater than in Sham (Fig. 4). These data revealed that Grafted NHPs, at 9 months post-surgery, had increased 11C-DTBZ binding in both putamina, indicative of functional DAN at the transplanted site, similar to changes in clinical trials using DAN from induced pluripotent stem cells [29].

L225

-          The authors should add figure references each time they are specifically referring to data in Figure 5.

We added the next information:

(Fig. 5, upper panels, Grafted 1 and 2).

(Fig. 5 bottom panels, Grafted 1 and 2).

L236

-          The H&E staining evidencing ‘[apparent lack of] innate or adaptive immune response’ seems a superficial analysis. Could these be staining for microglial, macrophage markers etc to add weight to this statement.

We performed immunodetection for IBA1 in the tissue of the 3 animals. Unfortunately, the antibody used (WAKO, #19-19741) did not detect microglial cells in the brain of NHPs. To validate the use of this antibody in human cells, we differentiated hESCs to mesoderm and then to microglia cells, which were detected with the IBA1 antibody:

Detection of Iba-1 in microglia-like cells derived from hematopoietic induction of the human embryonic stem cells line H9, that expresses GFP.

Unfortunately, we do not have more tissue to perform immunohistochemistry. Hematoxylin and eosin staining may suggest histopathological changes such as lymphocytic aggregates (doi.org/10.1371/journal.pone.0032796), or reactive microglia (DOI: 10.1177/0192623310389621), which were not observed in any of the animals. These references have been added to line 248.

L243

-          Correct spelling of ‘grafted’

Changed.

L282

-          Revise these summaries in the context of the earlier review comments.

We modified this section to tone down our findings.

L287

-          ‘..of the same [32]’ is there a missing word here?

Changed “same” by “graft”.

L302

-          ‘functional recovery … was a accompanied’ I think you mean ‘was suggest by’ – it is a stretch to say that functional recovery has simply occurred.

Changed.

L311

-          In Grafted 2 – does it really seem likely that the ‘edema or cell death’ is occurring at any great level, as the authors allude to with this sentence, when considering the all the evidence in the study? Seems an unhelpful suggestion unless the authors feel it is a real possibility that has not been excluded by the histological and other data they later covered.

We added the next statement:

Besides, the traumatic brain injury, generated by the needle, can generate changes per se in MD (Lin et al., 2016).

L313

-          Should give further detail to the effect of ‘…phase 1 clinical trial for the severe congenital demyelinating condition, Pelizaeus-Merbacher disease’.

We added the suggested information.

L364

-          Did the authors mean to say ‘…compensate for the loss of endogenous DA neurons’ or something to this effect here?

It means that compensates for the loss of Dopamine (DA). It is now stated in the text.

L371

-          Could the authors also add something on what their findings might suggest for the use of FA as a biomarker in clinical trials?

We added the next statement:

Notably, the findings in FA changes support the use of DTI as a biomarker for the quantitative analysis of neuronal damage in PD in clinical trials, since its reduction is associated with a lower number of NDA in the substantia nigra (doi:10.1177/2058460121993477) of these patients and could have efficiency in diagnosis and prognosis.

Methods:

L438

-          The authors must give more detail on the bioinformatic methods used in the RNAseq analysis, as well as details of the quality control from this such as genome coverage etc.

We added the section Gene expression analysis in Methods.

Gene expression analysis

We removed adaptor sequences using cutadapt [61] and trimmomatic [62] and obtained 36-76 bp paired-end reads for each sample. The quality of raw sequenced reads was verified using FASTQC [63]. Subsequently, we used a previously reported pipeline for read mapping, transcript assembly, and expression estimation [64] and mapped sequenced reads to the human reference genome hg38 (https://www.gencodegenes.org/) by using TopHat v2.1.1 [65] with default parameters. The reads were assembled and mapped using Cufflinks v2.2.1 [66] and calculated the values of Fragments Per Kilobase of transcript per million Mapped reads (FPKM) for all annotated genes and transcripts.

Sample

Reads PF

Paired Reads (Trimmomatic)

% Trim

RNA STAR

% STAR

RmDup

RmDup Aprox

RNA STAR Count Unmapped

RNA STAR Count Mapped

RNA STAR Count %

RNA STAR Count ALL %

Day0-r1

2,975,881

2,828,933

95.0%

2,510,778

88.7%

92.5%

 2,322,470

 557,997

1,952,781

77.78%

65.6%

Day0-r2

3,349,069

3,199,515

95.5%

2,847,575

89.0%

92.3%

 2,628,312

 637,844

2,209,731

77.60%

65.9%

Day0-r3

2,927,902

2,799,662

95.6%

2,471,487

88.2%

92.6%

 2,288,597

 552,669

1,918,818

77.64%

65.5%

Day14-r1

2,639,118

2,506,396

94.9%

2,212,330

88.2%

92.5%

 2,046,405

 474,216

1,738,114

78.56%

65.8%

Day14-r2

2,831,743

2,700,318

95.3%

2,401,428

88.9%

92.4%

 2,218,919

 498,600

1,902,828

79.24%

67.2%

Day14-r3

2,768,272

2,639,168

95.3%

2,348,184

88.9%

92.5%

 2,172,070

 497,287

1,850,897

78.82%

66.8%

Day14-r4

3,185,871

3,035,967

95.2%

2,709,484

89.2%

91.7%

 2,484,597

 569,966

2,139,518

78.96%

67.1%

Day28-r1

1,624,658

1,549,005

95.3%

1,393,858

89.9%

96.1%

 1,339,498

 255,928

1,137,930

81.64%

70.0%

Day28-r2

2,593,374

2,461,511

94.9%

2,214,291

89.9%

94.7%

 2,096,934

 412,712

1,801,579

81.36%

69.4%

The gene annotation was obtained from GENCODE v29 (https://www.gencodegenes.org/) and used HTSeq [70] to calculate read counts for annotated genes. Next, we performed a pairwise comparison of read counts implementing DESeq2 [71]. Genes were labeled as differentially expressed (DEG) if at least one of the replicates in the comparison had FPKM ≥ 1, and normalized count FC > 4 with an FDR < 0.05.

We obtained temporal gene expression profiles of DEGs using hierarchical clustering. In this unsupervised clustering method, we implemented Ward’s linkage algorithm using the Euclidean distance matrix of log2-transformed FPKM values of DEGs.

L524

-          ‘Sex does not alter this model 3,4 and two…’. Are 3 and 4 supposed to be references? They are not formatted as such.

We changed the format of these references.

Figures

L873

-          The image quality in figures 1C-E is too low. Could the authors please check that the images they are submitting are of a high standard and indicate if the issue is at their end or between the journal and the reviewers so this can be resolved.

We checked the images and improved the resolution.

L877

-          In Figure 1A, it is unclear what comparisons the indicated statistical differences refer to. The statistical method used is not indicated in figure legend, and it appears that multiple types of analysis were used. The authors must clarify this.

One-way ANOVA followed by Tukey’s test is added in Figure legend 1A,F and H

-          It is unclear what is meant by ‘independent experiments’. Are these independent differentiations? Parallel differentiations split from the same originator population? More detail is needed to assess exactly how independent these are from each other. This comment goes for other figure legends where this phrase is used.

Independent experiments are independent differentiations and now is enlightened in methods in Statistical analysis section. n values are from independent experiments (independent differentiations).

-          With these resolved I think the figure is satisfactory to show the point, but I wonder if normalizing only to GAPDH is robust; as, while stable within a particular line, GAPDH can change considerably in expression between cells of different lineages. The authors should consider using several ‘normal expression’ genes to counter this.

Thanks for the observation, which is very pertinent. At the same time also performed reactions with β-Actin as a housekeeping gene (data not shown) with very similar results.

L881

-          The authors should clarify how these genes were selected for display in this figure. Presumably they ae ‘top hits’ of some kind, but there should be detail of where they came and from what analysis. Some of this detail may be better placed in the methods, but it should be somewhere.

Part of the list of genes reported by Kriks et al, 2011 and is mentioned in methods in RNA-seq section.

L884

-          The authors should detail what is  meant by ‘quantitative co-expression analysis’, with summary of numbers of cells used, replicates etc.

Mean ± SEM; *p< 0.05, **p < 0.01, n=5 independent experiments.

L892

-          Please clarify in the figure legend for figure 1H that DA concentrations were measured by HPLC.

Legend is now clarified.

L897

-          In figure 2D, non-standardization of y axis limits within each of the motor task results impairs interpretation, the authors should correct his.

Y axis was standardized

L930

-          I am not experienced with these types of analysis and perhaps it is standard, but ‘percentage reduction (expressed as a percentage of Control)’ is a confusing metric to me. Suggest changing simply to % of control, with sham having 65%, Grafted 1 having 75%, Grafted 2 having 71% etc, or simply showing the PPOR ratios with SEM (as in the table) in a bar chart – I think it communicates the exact information.

We thank for this suggestion; we modified the figure accordingly.

Supplement

Supp. Fig 1D:

-          The authors do not detail the histological methods used to obtain these images. They must include this in the methods or at least in the supplementary figure legend. Additionally, the authors should help the reader by clarification on the relationship between the H&E staining in the upper panel and the fluorescence imaging in the lower panel. Were these from the same sample, adjacent sections etc?

The details were added at the end of Teratoma formation assay section.

“Euthanasia was performed (sodium thiopental, 21 mg/kg, i.p.) and mice were infused intracardially with PFA when tumors reached 4-5 mm. Teratomas were dissected and 5 mm slices were obtained, some were stained with hematoxylin and eosin and other adjacent slices were observed under the fluorescence microscope”. The legend of Supplementary Fig. 1D was modified.

Supp. Fig. 4

-          The authors title the figure legend that FA and MD decrease and increase, respectively. But no statistical justification is given for this sentence.

We changed it with

“MPTP intoxication of three NHPs diminishes Fractional Anisotropy (FA) and increases Mean Diffusivity (MD) in the putamen compared to healthy subjects”.

-          The authors should indicate the individual mice within each set, so that it is shown which are on the left side and right side and how these relate. This can be done by changing point shapes accordingly.

Thanks for the suggestion, colored circles now indicate the value for individual animals for Control and MPTP.

Round 2

Reviewer 1 Report

Comments and Suggestions for Authors Although there are some improvements in this manuscript, I still feel that the presentation of data and statistics is still misleading. The authors acknowledge in the cover letter that the measurements in the single sham monkey cannot be reasonably compared with that in 2 MPTP-treated monkeys.  The authors correctly state in their cover letter: "We agree that the most important statistical analysis would have been between groups, which unfortunately could not be performed given the n".  However the manuscript conclusion deviates from this: "The findings of this study support, at the molecular, biochemical, imaging, and behavioral levels, the notion that stem cells differentiated to DAN can be grafted into MPTP NHP PD model, where they integrate, survive, and release DA which results in behavioral  change".   This sentence directly links the behavioral changes with the transplanted cells, which implies a statistical difference from controls. While one can appreciate the authors' desire to paint the data in the most interesting light, it is imperative that data are not oversold as this would mislead readers who are not very familiar with the research field or biostatistics. 

Reviewer 3 Report

Comments and Suggestions for Authors

The manuscript of López-Ornelas and colleagues encompasses several complementary analyses of aspects of an experimental ES-derived neuron replacement therapy in non-human primates. There are several interesting new findings and, given the use of NHP animal models, this represents an informative exploration of a pre-clinical therapeutic approach. While limited in power, the study is comprehensive in content and does well to expand on studies of similar therapeutic interventions.

I appreciate the work done by the authors in the revision of this manuscript, which is well-received and I believe has improved several areas of the manuscript. I do note though that a few of the changes either introduce new issues or did not sufficiently resolve the original issue raised.

Figure 2F

- For the new panel of analyses the authors have added showing the single-case analysis the authors should provide the name of the statistical test used.

- In the new panel for “Reaching”, there is not a significant difference shown between Sham and Grafted 2. Given the bars of the SEM shown, is this correct? If this is because the bars of the SEMs are distinct from the single-case analysis, then I would question if it is appropriate to show SEM bars alongside, rather than an appropriate measure of variability within the scope of the single-case analysis.

Figure 4

- The authors have modified the y axis in the final panel to measure % relative to control, but by capping the axis at 74% this obscures the fact that the recovery is still far short of being comparable to controls and is only slightly different to Sham. While use of a y limit at 62% is acceptable as they have established the baseline with the Sham animal, they should show the change relative to control overall and therefore the y axis should reach 100% and the Control should be added to the bars for clarity.

L194

- I must stress that the authors did not mention, in the body of text to which I refer, that decreased FA and increase MD were not shown to occur with statistical confidence. While it is mentioned in the legend of the supplementary figure this is not sufficient and the authors must clearly state the negative outcome of the statistical test in the body text if they are to cite that there are potential changes in these measures.

- Moreover, I underline that failure to meet statistical thresholds is an indicator of lack of confidence that there is any difference at all between case and control, and not an indicator of difference to a lesser degree as the authors seem to imply. Therefore it is essential to highlight clearly in the citing body text that this change is not significant.

- Use of a 95% confidence interval of the degree of change would also be a fair representation and may better match the author’s objective with this sentence.

L392

[language suggestion]

- Suggest starting paragraph with ‘A recent clinical trial used imaging…’ or similar.
